# Symbiotic bacteria-dependent expansion of MR1-reactive T cells causes autoimmunity in the absence of Bcl11b

Kensuke Shibata[1,2,3], Chihiro Motozono [3,4], Masamichi Nagae [3,5], Takashi Shimizu[3], Eri Ishikawa[3,5], Daisuke Motooka[6], Daisuke Okuzaki [7,8], Yoshihiro Izumi [9], Masatomo Takahashi [9], Nao Fujimori[10], James B. Wing [11,12], Takahide Hayano[13], Yoshiyuki Asai[13], Takeshi Bamba [9], Yoshihiro Ogawa [10,14,15], Makoto Furutani-Seiki [16], Mutsunori Shirai[1] & Sho Yamasaki [3,5,17,18] ✉

MHC class I-related protein 1 (MR1) is a metabolite-presenting molecule that restricts MR1-reactive T cells including mucosal-associated invariant T (MAIT) cells. In contrast to MAIT cells, the function of other MR1-restricted T cell subsets is largely unknown. Here, we report that mice in which a T cell-specific transcription factor, B-cell lymphoma/leukemia 11B (Bcl11b), was ablated in immature thymocytes (*Bcl11b*^ΔiThy mice) develop chronic inflammation. *Bcl11b*^ΔiThy mice lack conventional T cells and MAIT cells, whereas CD4$^+$IL-18R$^+$ αβ T cells expressing skewed *Traj33* (Jα33)$^+$ T cell receptors (TCR) accumulate in the periphery, which are necessary and sufficient for the pathogenesis. The disorders observed in *Bcl11b*^ΔiThy mice are ameliorated by MR1-deficiency, transfer of conventional T cells, or germ-free conditions. We further show the crystal structure of the TCR expressed by *Traj33*$^+$ T cells expanded in *Bcl11b*^ΔiThy mice. Overall, we establish that MR1-reactive T cells have pathogenic potential.

Unlike conventional T cells, unconventional T cells such as invariant NKT (iNKT) cells and mucosal-associated invariant T (MAIT) cells display restricted TCR repertoires characterized by the expression of invariant TCRα chains. For example, MAIT cells express TRAV1-2-TRAJ33 in humans and the orthologous Trav1-Traj33 in mice. These TCRs recognize microbial riboflavin precursors such as 5-(2-oxopropylideneamino)−6-D-ribitylaminouracil (5-OP-RU) presented by MHC class I-related 1 (MR1) proteins[1,2]. Upon bacterial infection, MAIT cells are activated rapidly and elicit protective responses such as the production of IFNγ and IL-17A or cytotoxicity[3]. Although typical type 2 MAIT cells have not been reported, MAIT cells capable of producing type 2 cytokines were detected[4]. Recent reports have, in addition, identified non-canonical TRAV1$^{neg}$ MR1-reactive T cells that do not express the TCR or markers characteristic of MAIT cells[5]. The physiological/pathological functions of these non-canonical MR1-reactive T cells have not yet been fully elucidated.

Bcl11b is a T cell-specific transcription factor essential for the transition from double-negative (DN) to double-positive (DP) thymocytes[6–9]. Genetic deletion of Bcl11b in immature thymocytes (before the DP stage) (*Bcl11b*^flox/flox × *Rag1*^Cre/+; referred to as *Bcl11b*^ΔiThy mice) impaired the development of most T cell subsets. However, we previously observed that unconventional T cell subsets including Vγ1-bearing γδ T cells, develop from DN thymocytes in *Bcl11b*^ΔiThy mice[10]. Other similar "early developing" populations of lymphocytes have been recently identified[9,11,12]. We have herein used our *Bcl11b*^ΔiThy mice to characterize a population of these additional, as yet unappreciated, unconventional cells that develop from early thymocytes. In the present study, we demonstrate that Trav1$^{neg}$ atypical MR1-reactive T cells expand in *Bcl11b*^ΔiThy mice and acquire pathogenic potential, causing hypergammaglobulinemia and chronic inflammation in a process dependent on symbiotic bacteria.

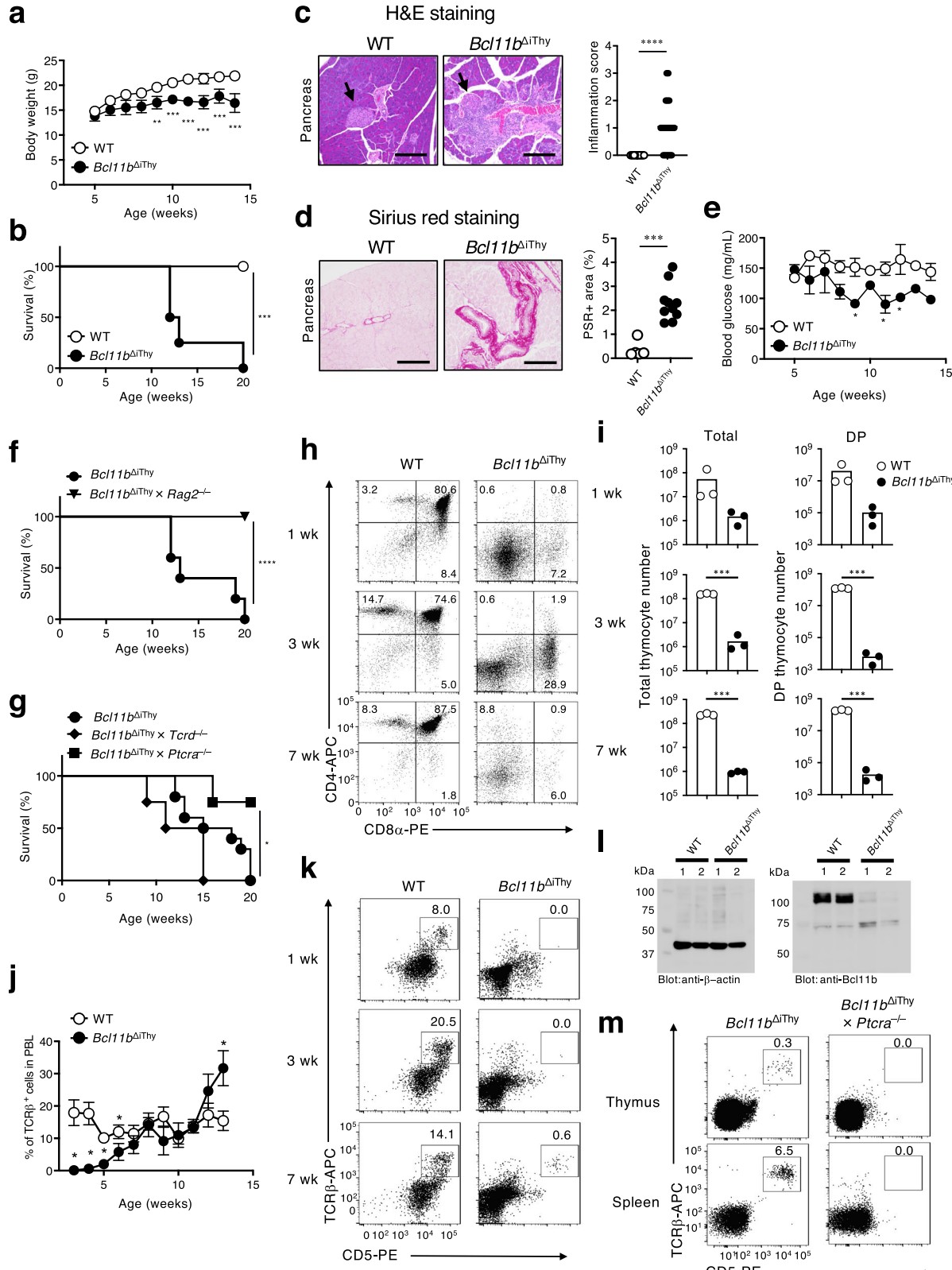

## Results

### *Bcl11b*^ΔiThy mice develop chronic inflammation

*Bcl11b*^flox/flox × *Rag1*^Cre/+ (*Bcl11b*^ΔiThy mice) were normal at birth[10], but they did not gain weight at around 7 weeks of age and died by 15–20 weeks (Fig. 1a, b). Histological analysis of organs from *Bcl11b*^ΔiThy mice showed extensive infiltration of inflammatory cells into the periductal areas of the pancreas and disruption of acinar architecture (Fig. 1c). *Bcl11b*^ΔiThy

mice also exhibited periductal fibrosis and atrophy of the exocrine portion of the pancreas (Fig. 1d). Serum lipase activity was elevated as inflammation progressed, indicating that *Bcl11b*^ΔiThy mice suffered severe pancreatic damage (Supplementary Fig. 1a). However, *Bcl11b*^ΔiThy mice did not develop diabetes (Fig. 1e), consistent with the observation that islets containing β cells were not damaged (Fig. 1c). In addition, we observed thickened lamina propria and shortened large

**Fig. 1 | *Bcl11b*$^{\Delta iThy}$ mice spontaneously develop chronic inflammation. a, b** Age-related changes of body weights (**a**) and survival rates (**b**) of WT ($n = 8$) and *Bcl11b*$^{\Delta iThy}$ mice ($n = 8$). Asterisks indicate statistical significance determined by Two-way ANOVA (**a**) (**$p < 0.01$, ***$p < 0.001$) and by logrank tests (***$p = 0.0002$) (**b**). **c, d** H&E staining (**c**) and sirius-red staining (**d**) of pancreatic sections from WT and *Bcl11b*$^{\Delta iThy}$ mice (15 weeks old). Scale bar shows 200 μm. **c** An arrow head indicates islet. ****$p < 0.0001$ by unpaired two-tailed Student's *t*-tests. **d** Quantified data of inflammation scores based on histological analysis of pancreas of WT and *Bcl11b*$^{\Delta iThy}$ mice (15–20 weeks old) was shown on the right panel. ***$p = 0.0001$ by unpaired two-tailed Student's t-tests. **e** Blood glucose levels in WT and *Bcl11b*$^{\Delta iThy}$ mice ($n = 3$ biologically independent animals) at different ages. Data are expressed as mean ± SEM. *$p < 0.05$ by two-way ANOVA tests. **f, g** Survival of the indicated strains of mice (*Bcl11b*$^{\Delta iThy}$ mice, $n = 6$; *Bcl11b*$^{\Delta iThy}$ × *Rag2*$^{-/-}$ mice, $n = 10$; *Bcl11b*$^{\Delta iThy}$ mice, $n = 10$; *Bcl11b*$^{\Delta iThy}$ × *Tcrd*$^{-/-}$ mice, $n = 8$; *Bcl11b*$^{\Delta iThy}$ × *Ptcra*$^{-/-}$ mice, $n = 8$) was monitored for 20 weeks. ****$p < 0.0001$, *$p = 0.0149$ by logrank tests. **h, i, k, m** Analysis of thymocytes (**h, i, k, m**) and splenocytes (**m**) in indicated mice (**h, i, k**: 1, 3, 7 weeks old, m: 9 weeks old). Representative dot plots are shown after gating on total lymphocytes (**h, k, m**). **i** Bar graphs show absolute numbers of total (left) and DP (right) thymocytes. ***$p < 0.001$ by unpaired two-tailed Student's *t*-test. **j** αβ T cell frequencies in the peripheral blood lymphocytes (PBL) of WT and *Bcl11b*$^{\Delta iThy}$ mice (at least $n = 3$ in each group) over time. Data are expressed as mean ± SEM. *$p < 0.05$ by two-way ANOVA tests. **l** Western blot analysis of Bcl11b expression in sorted αβ T cells from two individual WT and *Bcl11b*$^{\Delta iThy}$ mice. β-actin: 42 kDa, Bcl11b: 96 kDa. Gating strategies are shown in Supplementary Fig. 6a for Fig. 1h, k and m (thymocytes), and Supplementary Fig. 6b for Fig. 1m (splenocytes). **a, e, i, j** Data are representative of three independent experiments. **b, f, g** Data are combined from at least two independent experiments.

intestines in *Bcl11b*$^{\Delta iThy}$ mice (Supplementary Fig. 1b, c). We also observed inflammation albeit less frequently in other organs such as the lung and lachrymal gland (Supplementary Fig. 1d). Thus, *Bcl11b*$^{\Delta iThy}$ mice develop spontaneous inflammation.

The inflammation and decreased survival in *Bcl11b*$^{\Delta iThy}$ mice were absent on a Rag2-deficient background (Fig. 1f, Supplementary Fig. 1e); this suggests that T lymphocytes are required for these defects since Bcl11b is not expressed in B cells[8,13]. γδ T cells, which are present in *Bcl11b*$^{\Delta iThy}$ mice[10], are not necessary for disease progression as *Bcl11b*$^{\Delta iThy}$ × *Tcrd*$^{-/-}$ mice still displayed this lethal disorder (Fig. 1g); in contrast, survival was significantly prolonged on a pre-TCRα-deficient background (*Bcl11b*$^{\Delta iThy}$ × *Ptcra*$^{-/-}$ mice) in which αβ T cells are absent (Fig. 1g). Notably, despite the DN2 arrest phenotype[10] (Supplementary Fig. 2a, Fig. 1h, i), a few CD5$^+$ αβTCR$^+$ thymocytes were observed in 7-week-old *Bcl11b*$^{\Delta iThy}$ mice (Fig. 1k). In the periphery, αβTCR$^+$ cells were extremely rare at an early age; however, the cell number gradually increased with age in *Bcl11b*$^{\Delta iThy}$ mice and exceeded that of WT mice (Fig. 1j). It is unlikely that this accumulation of mature T cells is due to incomplete deletion of Bcl11b in those cells, as Bcl11b protein could not be detected in peripheral αβ T cells from *Bcl11b*$^{\Delta iThy}$ mice (Fig. 1l). In the absence of preTCRα chains (*Bcl11b*$^{\Delta iThy}$ × *Ptcra*$^{-/-}$ mice), such peripheral αβ T cells were eliminated (Fig. 1m), indicating that these T cells must undergo pre-TCR-mediated β-selection. Furthermore, adoptive transfer of T cells from *Bcl11b*$^{\Delta iThy}$ mice into CD3ε-deficient mice was sufficient to induce the disorders (Supplementary Fig. 2b). These data suggest that αβ T cells that develop in the absence of Bcl11b possess pathogenic potential.

### Pathogenic contribution of non-canonical MR1-reactive T cells

Most αβ T cells in *Bcl11b*$^{\Delta iThy}$ mice displayed a CD44$^{high}$CD62L$^{low}$IL-18R$^{high}$ phenotype, implying that they include unconventional T cells (Fig. 2a)[3]. However, CD1d-restricted iNKT cells were not detected in either the thymus or the periphery of *Bcl11b*$^{\Delta iThy}$ mice (Fig. 2b, c, Supplementary Fig. 2c, d). To evaluate the contribution of MR1-reactive αβ T cells to these disorders, *Bcl11b*$^{\Delta iThy}$ mice were crossed with MR1-deficient mice (*Bcl11b*$^{\Delta iThy}$ × *Mr1*$^{-/-}$ mice). Pancreatitis was ameliorated in the resulting *Bcl11b*$^{\Delta iThy}$ × *Mr1*$^{-/-}$ mice (Fig. 2e) and they survived longer (Fig. 2d). However, neither MR1 tetramer-positive (5-OP-RU-tet$^+$) MAIT cells nor the canonical TCRα repertoire of MAIT cells (Trav1-Traj33) were detected in the spleens of *Bcl11b*$^{\Delta iThy}$ mice (Fig. 2b, c). Nevertheless, the proportion of CD44$^{high}$ IL-18R$^{high}$ αβ T cells observed in *Bcl11b*$^{\Delta iThy}$ mice was reduced to a level comparable to that of WT mice in *Bcl11b*$^{\Delta iThy}$ × *Mr1*$^{-/-}$ mice (Fig. 2f). The T cells that accumulated in *Bcl11b*$^{\Delta iThy}$ mice did not express PLZF, which is a Bcl11b-dependent transcription factor characteristic of MAIT cells and iNKT cells (Supplementary Fig. 2e)[14], confirming that they are not MAIT cells. Thus, the expansion of non-canonical MR1-reactive αβ T cells is associated with the development of chronic inflammation in *Bcl11b*$^{\Delta iThy}$ mice.

### Correction of T cell insufficiency ameliorates chronic inflammation

*Bcl11b*$^{\Delta iThy}$ mice lack most conventional αβ T cells[10]. We therefore speculated that this T cell imbalance may allow the development/expansion of pathogenic innate-type T cells. After the transfer of bone marrow (BM) cells from *Bcl11b*$^{\Delta iThy}$ mice into CD3ε-deficient mice, recipient mice experienced body weight loss in a lymphocyte-dependent (Rag1-dependent) manner (Fig. 2g). However, when BM cells from *Bcl11b*$^{\Delta iThy}$ and WT mice were mixed at a 1:1 ratio and the combination transferred, the body weights of recipient mice remained normal (Fig. 2g). Consistent with this, expansion of pathogenic CD4$^+$ *Bcl11b*$^{\Delta iThy}$ T cells (Supplementary Fig. 2b) was constrained in the presence of WT cells (Fig. 2h–j), possibly due to the survival disadvantages[15]. Furthermore, chronic inflammation induced by *Bcl11b*$^{\Delta iThy}$ BM cells was eliminated by co-transfer with WT BM cells (Fig. 2k, Supplementary Fig. 2f). The effect of the presence of balanced T cells was also confirmed by using non-irradiated *Bcl11b*$^{\Delta iThy}$ recipient mice, as transfer of WT T cells limited the expansion of donor T cells (Fig. 2m), inflammations (Fig. 2k, Supplementary Fig. 2f) and fatal disorder independently of regulatory T cells (Treg) (Fig. 2l). These results suggest that the imbalance in T cells observed in the *Bcl11b*$^{\Delta iThy}$ genetic environment contributes to the pathogenesis.

### MR1-dependent expansion of Traj33-bearing T cells in *Bcl11b*$^{\Delta iThy}$ mice

To characterize the profiles and clonotypes of the pathogenic T cells at a clonal level, we performed single-cell-based TCR and RNA-sequencing (scTCR-RNAseq). In *Bcl11b*$^{\Delta iThy}$ mice, the TCR repertoire was highly skewed with small number of clones dominating (Fig. 3a). The most frequent clone in *Bcl11b*$^{\Delta iThy}$ mice accounted for 15% of total αβ T cells and expressed a *Trav7-6-Traj33/Trbv13-2-Trbj1-1* TCR (designated as clonotype #**1**) with identical CDR3α and CDR3β sequences (Fig. 3a–c). Cells expressing clonotype #**1** occupied 87% of all *Traj33*$^+$ T cells in *Bcl11b*$^{\Delta iThy}$ mice (Fig. 3c, d, left panels). Furthermore, the TCRα and β chains of clonotype #**1** preferentially paired with each other (Supplementary Fig. 3a, b). Although the Vα usage of clonotype #**1** (Trav7-6) differed from the canonical Trav1$^+$ MAIT TCR, the Traj33 Jα, which provides the critical amino acids for MR1 contact, was identical to typical MAIT cells[16] (Fig. 3b, Supplementary Fig. 3c, d). We therefore solved the crystal structure of the apo form of the clonotype #**1** TCR (Fig. 3e) and superimposed this structure onto the complex structure of the MR1-6-formylpterin (6FP)-MAIT TCR[17]. The comparison suggested that the Tyr residue in the SNYQ sequence, particularly Y$^{97}$, provided by the Jα33 chain, was located in a similar but different position to the Y$^{95}$ of canonical MAIT TCR, suggesting that it pointed toward the ligand binding pocket of MR1 in a slightly different manner[18] (Fig. 3f, g, Supplementary Fig. 3e). Indeed, the frequency of *Traj33*$^+$ T cells, including those expressing clonotype #**1**, was dramatically decreased on an MR1-deficient background (Fig. 3c, d, right

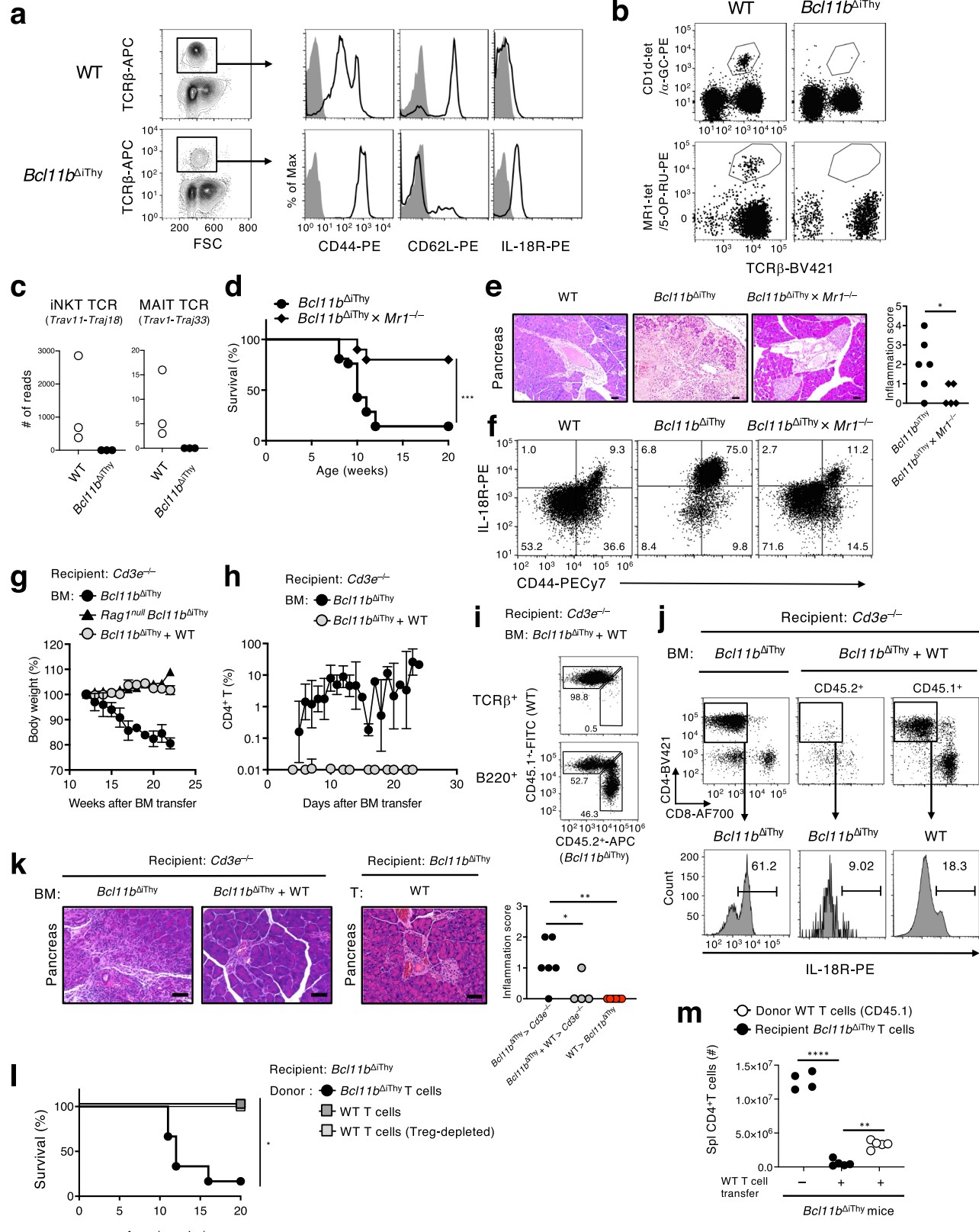

panels), indicating that MR1 promotes the development and/or expansion of *Traj33*[+] non-MAIT cells in *Bcl11b*[ΔiThy] mice.

Simultaneous single-cell RNA-seq analysis implied that clonotype #**1** was detected in different clusters including *Mki67*[high] proliferating cells, and *Mki67*[neg] cells expressing *Il4*, *Ifng* or *Gzmb* (Fig. 3h). No IL-17A-positive cluster was detected. Pseudotime analysis of clonotype #**1**

allowed us to visualize three distinct differentiation pathways when *Mki67*[high] cells were defined as a root for the analysis (Fig. 3i, j). Along the pseudotime dimension, the proliferative ability identified by *Mki67* expression gradually decreased to the end, suggesting that functional differentiation occurred after proliferation (Fig. 3k). One of the three computationally predicted lineages (Lin 1) was characterized by *Il4*

**Fig. 2 | MR1-reactive αβ T cells are pathogenic in *Bcl11b*^ΔiThy mice. a** Histograms of indicated surface markers (black line) on splenic CD4⁺ αβ T cells from indicated mice (10 weeks old) compared with isotype control (filled gray). **b** α-GC-loaded mCD1d tetramer⁺ cells and 5-OP-RU-loaded mMR1 tetramer⁺ cells in lymphocytes. **c** Each dot represents read count derived from transcript of *Trav11-Traj18* and *Trav1-Traj33* from the spleens of WT and *Bcl11b*^ΔiThy mice (*n* = 3 biologically independent animals). **d** Survival of *Bcl11b*^ΔiThy mice (*n* = 12) and *Bcl11b*^ΔiThy × *Mr1*⁻/⁻ mice (*n* = 9). ***p* = 0.0010 by logrank tests. **e, f** H&E staining of sections of pancreas with quantified data (**e**) and IL-18R and CD44 expressions on splenic αβ T cells (**f**) from indicated mice (15 weeks old). **e** Scale bar shows 50 μm. **p* = 0.0420 by unpaired two-tailed Student's *t*-tests. **g–k** BM cells from *Bcl11b*^ΔiThy mice (closed circle) or *Rag1*^null *Bcl11b*^ΔiThy (*Bcl11b*^flox/flox × *Rag1*^Cre/Cre) mice (closed triangle) were transferred into *Cd3e*⁻/⁻ mice. *Bcl11b*^ΔiThy BM cells were mixed with WT BM cells at 1:1 ratio before transfer (gray circle) (*n* = 3–5 biologically independent animals in each group). After transfer, body weights (**g**) and CD4⁺ αβ T cell frequencies (**h**) were monitored. **g, h** Data are expressed as mean ± SEM. **i–k** 26 weeks after transfer, frequencies of

αβ T cells and B cells (**i**), CD4 and CD8α (upper) or IL-18R (lower) expressions (**j**), H&E staining of sections of pancreas with quantified data (**k**) in recipient mice were analyzed. **k** Scale bar shows 60 μm. **p* = 0.0316, ***p* = 0.0089 by unpaired two-tailed Student's *t*-test. **l, m** Whole T (dense gray square) or Treg-depleted T cells (light gray square) from WT mice or whole T cells from *Bcl11b*^ΔiThy mice (black circle) were transferred into 3-week-old *Bcl11b*^ΔiThy mice. **l** Survival of *Bcl11b*^ΔiThy mice receiving *Bcl11b*^ΔiThy T cells (*n* = 6), WT T cells (*n* = 5) and Treg-depleted T cells (*n* = 5). **p* = 0.032 by logrank tests between mice receiving WT T cells and Treg-depleted T cells. **m** 16 weeks after transfer of the donor WT splenic T cells (CD45.1) to recipient *Bcl11b*^ΔiThy mice (3 weeks old), the numbers of splenic CD4⁺ T cells were analyzed. ****p* < 0.0001 by unpaired two-tailed Student's *t*-tests or ***p* = 0.0019 by paired two-tailed Student's *t*-tests. Data are representative of (**a, b, f**) three and (**c, i, j**) two independent experiments. Gating strategies are shown in Supplementary Fig. 6b for Fig. 2b, and Supplementary Fig. 6c for Fig. 2f, i and j. Data are combined from three (**d**) or two (**e, g, h, k, l, m**) independent experiments.

expression and gradually acquired expression of T_FH marker genes such as *Bcl6, Cxcr5, Pdcd1, Il21, Sostdc1* and *Rgs16*[19–21] (Fig. 3k, Supplementary Fig. 3g). Another lineage (Lin 2) (Fig. 3j) overlapped with a CTL-like cluster (Supplementary Fig. 3h). Importantly, the T cells bearing *Trav7-6-Traj33* was also detected in other individual *Bcl11b*^ΔiThy mice and even in WT mice at a low frequency (Supplementary Fig. 3f), suggesting that these T cells are not arisen sporadically and aberrantly under *Bcl11b*-deficient conditions.

Thus, *Traj33*⁺ T cells bearing non-MAIT TCRs are clonally expanded in the presence of MR1 and proceed through functional differentiation in *Bcl11b*^ΔiThy mice.

### Elevated serum immunoglobulin and autoantibodies in *Bcl11b*^ΔiThy mice

Consistent with the expansion of T_FH-like cells expressing IL-4, IgG levels in the serum of *Bcl11b*^ΔiThy mice began to increase at about the same age as when αβ T cells began to be detected (Fig. 4a). All IgG subclasses were significantly higher in *Bcl11b*^ΔiThy mice (Fig. 4b). Sera from individual *Bcl11b*^ΔiThy mice reacted to lysates from mouse pancreas, suggesting the presence of autoantibodies (Fig. 4c). Furthermore, IgG1 antibodies were deposited around pancreatic ductal sites in *Bcl11b*^ΔiThy mice; this was not observed on an MR1-deficient background (Fig. 4d). Both the number of IgG1⁺ B cells and IgG1 production were increased in the peripheral blood of *Bcl11b*^ΔiThy mice in the presence of MR1 (Fig. 4e, f) and pre-TCRα (Supplementary Fig. 4a). These data suggest that *Bcl11b*^ΔiThy mice develop hyperimmunoglobulinemia including autoantibodies in an MR1-restricted T cell-dependent manner. Indeed, preventing the expansion of such pathogenic T cells by adoptive transfer of WT T cells recovered *Bcl11b*^ΔiThy mice from hyperimmunoglobulinemia (Supplementary Fig. 4b).

Consistent with the high IgG1 production in *Bcl11b*^ΔiThy mice, mass cytometry analysis showed increases in germinal center (GC) B cells and T_FH-like cells in *Bcl11b*^ΔiThy mice (Fig. 4g, Supplementary Fig. 4c, d). We also observed peanut agglutinin positive (PNA⁺) germinal center-like structures in the enlarged pancreatic lymph nodes of *Bcl11b*^ΔiThy mice, which was not observed in WT mice (Supplementary Fig. 4e). Consistent with these observations, CD4⁺ αβ T cells in *Bcl11b*^ΔiThy mice expressed CD40L (Fig. 4h) and produced IL-4 (Fig. 4i). IL-4-GFP reporter fluorescence was also detected in pancreatic LN T cells from *Bcl11b*^ΔiThy × *Il4*^GFP/+ mice (Fig. 4j). On either a B cell-deficient (μMT) or IL-4-deficient (*Il4*^GFP/GFP) background, *Bcl11b*^ΔiThy mice did not survive longer (Supplementary Fig. 4f). The inflamed pancreases of *Bcl11b*^ΔiThy mice were infiltrated with some IgG1⁺ B cells and CD4⁺ αβ T cells in an MR1-dependent manner (Supplementary Fig. 4g, h). These clinical features are similar to those in human autoimmune pancreatitis (AIP)[22], which has been characterized by the high levels of serum human IgG4 (a putative counterpart of mouse IgG1[23]), periductal pancreatitis, lymphocyte infiltration and steroid responsiveness[24]. Indeed, steroid

treatment ameliorated elevation of serum IgG and pancreatitis in *Bcl11b*^ΔiThy mice (Fig. 4k, l).

### Remission of pancreatitis in *Bcl11b*^ΔiThy mice under germ free condition

A canonical MAIT cell antigen, 5-OP-RU, is provided mainly by microbiota[1,25,26]. We thus assessed whether symbiotic microbes affect the pathogenic T cells and disorders observed in *Bcl11b*^ΔiThy mice by generating germ-free (GF) *Bcl11b*^ΔiThy mice. Under germ-free conditions, we did not observe weight loss (Fig. 5a), a survival defect (Fig. 5b), T cell expansion (Fig. 5c), pancreatitis (Fig. 5d) or hypergammaglobulinemia (Fig. 5e) in *Bcl11b*^ΔiThy mice. Furthermore, the number of peripheral αβ T cells expressing high levels of IL-18R was reduced in GF *Bcl11b*^ΔiThy mice compared with *Bcl11b*^ΔiThy mice under specific pathogen-free (SPF) conditions (Fig. 5f–h). We also performed scTCR-RNA-seq on T cells from GF *Bcl11b*^ΔiThy mice and did not detect clonotype #1 (Fig. 5i), although this clonotype accumulated in SPF *Bcl11b*^ΔiThy mice (Fig. 3c). These results suggest that symbiotic microbes are required for the fatal disorders in *Bcl11b*^ΔiThy mice. Indeed, like germ-free conditions, treatment with antibiotics also ameliorated chronic inflammation (Fig. 5j) and improved survival (Fig. 5k) in *Bcl11b*^ΔiThy mice.

### Clonotype #1 responds to fecal components

The above observations suggest that symbiotic microbes may provide an antigen(s) to clonotype #1 to support clonal expansion and/or activation. We thus reconstituted the expression of clonotype #1 TCRα (Trav7-6-Traj33) and TCRβ (Trbv13-2-Trbj1-1) in a TCRαβ-deficient reporter cell line (Fig. 6a). To identify antigenic components, fecal extracts from SPF mice were fractionated and tested for stimulatory activity towards reporter cells expressing the clonotype #1 TCR in the presence of MR1-expressing antigen-presenting cells. Fraction #54 was found to induce NFAT reporter activity in an MR1-dependent manner (Fig. 6b), as it was suppressed in the presence of Ac-6-FP (Fig. 6c). To determine the active component(s) in fraction #54, we analyzed these seventy fractions by non-targeted metabolomics[27]. Twenty-two candidates were selectively and reproducibly present in fraction #54 (Fig. 6d, Supplementary Fig. 5a, b). Among them, only one candidate (#54-6) had similar *m/z* value to the exact mass of a known metabolite (human metabolome database, https://hmdb.ca), riboflavin (RF) (Supplementary Fig. 5b). MS/MS analysis confirmed that this precursor ion is derived from RF (Fig. 6e). Indeed, synthetic RF stabilized the surface expression of MR1 (Fig. 6f), suggesting that RF specifically binds to MR1 as recently reported[28]. Although RF was not recognized by MAIT TCRs (Fig. 6g, blue)[28], it activated cells expressing the clonotype #1 (Fig. 6g, red) and this activation was blocked by both Ac-6-FP and anti-MR1 antibody (Fig. 6g, black). As predicted from its TCR sequence (Fig. 3b, Supplementary Fig. 3c), clonotype #1 did not react

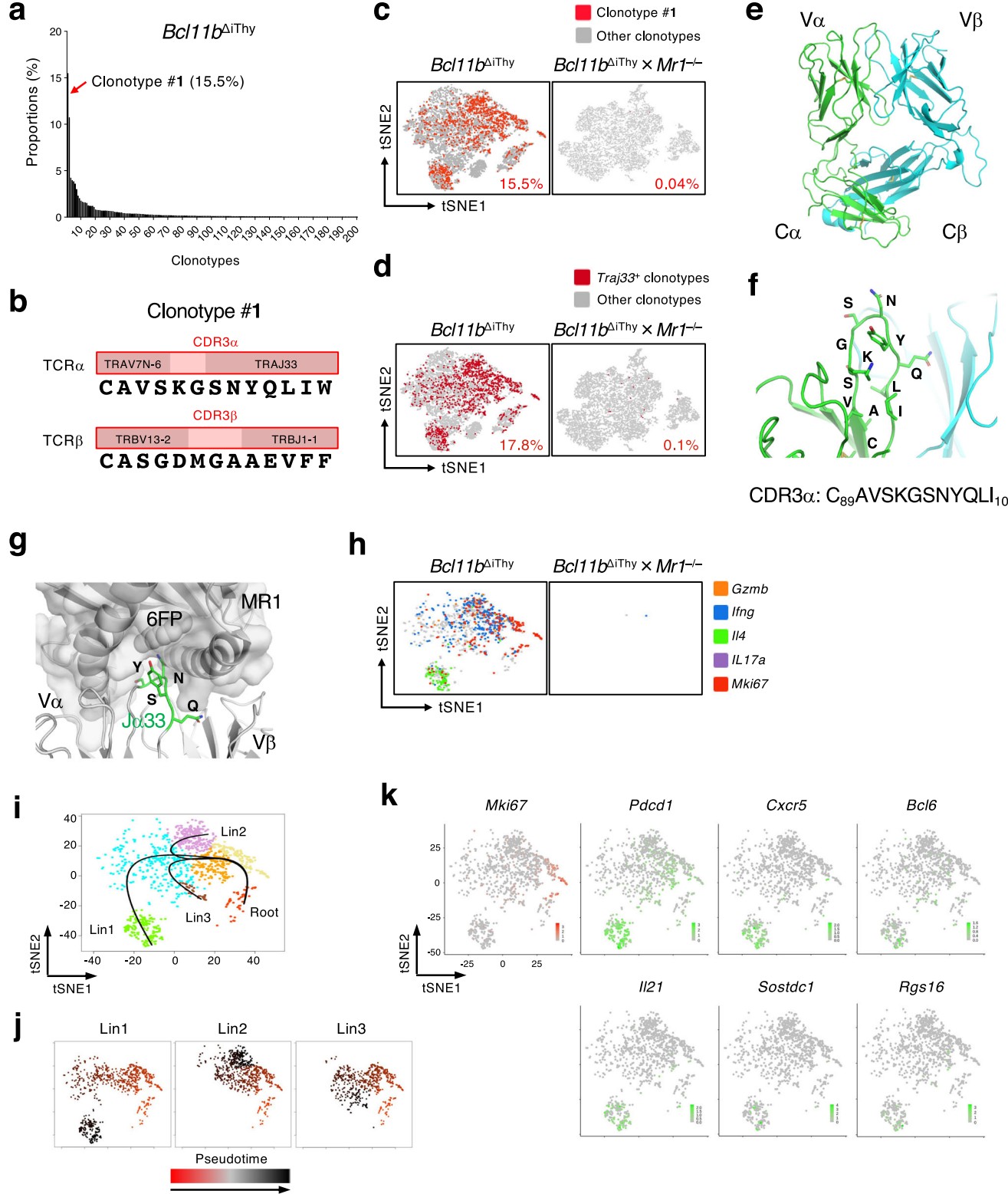

with an authentic MAIT cell antigen, 5-OP-RU (Fig. 6h). These results suggest that non-canonical MR1-reactive T cells recognize metabolites presented by MR1 that are distinct from MAIT cell antigens.

## Discussion

In the present study, we demonstrate that non-canonical MR1-reactive $\alpha\beta$ T cells acquire pathogenic potential through microbiota-dependent clonal expansion. RF metabolites, such as 5-OP-RU, are agonistic antigens for MAIT cells[2], whereas RF itself is a weak

antagonist of MAIT cells[28]. Considering its different action on expanded clonotypes in our study, RF might regulate MR1-reactive T cells as both an agonist and an antagonist depending on the TCR repertoire[28]. RF may contribute to the maintenance of MR1-reactive T cells in normal setting; however, excessive intake of RF might influence inflammation in particular T-imbalanced settings. It is also possible that, in addition to RF, the active fraction #54 contains an antigen(s) which has not yet been annotated (Fig. 6d and Supplementary Fig. 5b). Taken together with recent reports[28-32], more diverse metabolites than

**Fig. 3 | Identification of pathogenic T cells by single cell analysis. a–d, h** Sorted T cells from *Bcl11b*$^{\Delta iThy}$ mice or *Bcl11b*$^{\Delta iThy}$ × *Mr1*$^{-/-}$ mice (11 weeks old) were analyzed by single-cell TCR and transcriptome analysis. **a** Frequencies of the top 200 clones in T cells from *Bcl11b*$^{\Delta iThy}$ mice. **b** Junctional amino acid sequences of the TCRα and β chains of the top 1 clonotype (clonotype #**1**). **c, d** tSNE plots generated by data based on single cell transcriptome and TCR analysis of the indicated strains of mice; clonotype #**1** (red) or *Traj33*$^+$ clonotypes (wine red) and other clonotypes (gray) are shown. **e** Crystal structure of clone #**1** TCRαβ ectodomain. **f** Closeup view of CDR3α regions. **g** Putative interaction between Traj33 and MR1. Crystal structure of MAIT TCR-MR1-6-FP ternary complex is superposed onto clone #**1** TCRαβ heterodimer. The positions of Traj33 are highlighted (green). **h** Colored dots indicate clonotype #**1** with significantly high expression levels of indicated genes. **i** Seven clusters of clonotype #**1** cells after analysis by mclust. **j** Three distinct lineages (Lin1-3) of clonotype #**1** cells after analysis of slingshot. The pseudotime can be arbitrarily "stretched" by factors such as the magnitude of differential expression or the density of cells, depending on the algorithm for each lineage. The absolute values of the pseudotime in a lineage make little sense. Hence, the magnitude of the pseudotime has little comparability across lineages. In each panel, the values of pseudotime were normalized to 0–100 corresponding to orange-red and black, respectively. **k** Heat map graphs of expressions of indicated genes involved in T$_{FH}$ cell differentiation in clonotype #**1** cells. Source data are provided as Source Data files; Supplementary_code.text and Supplementary_data.RData.

previously demonstrated may regulate various mouse and human MR1-reactive T cells.

An important question arising is whether pathogenic T cells, such as clonotype #**1**, are also detected in Bcl11b-sufficient mice. CDR3-matched *Trav7-6-Traj33* TCRs were identified by bulk TCR sequencing of WT mice; however, such cells exist at low frequency (Supplementary Fig. 3f) and are not pathogenic. Thus, a normal environment in which innate T cell subsets are balanced is likely to restrict excessive expansion of such clonotypes, thus preventing pathogenic conversion[10,33]. Development of probes, such as specific tetramers or clonotypic mAbs, would enable us to monitor the status of these clonotypes during various experimental settings.

It should be noted that Bcl11b is critical for the development and function of Treg cells, since CD4-Cre-driven Bcl11b-deficient mice develop inflammatory diseases due to the dysregulated activation of effector Th17 cells via defective Treg function[34]. This scenario, however, may not explain the symptoms observed in Rag1-Cre-driven Bcl11b-deficient (*Bcl11b*$^{\Delta iThy}$) mice, as the correction of T-imbalance could suppress the disease in the absence of Treg (Fig. 2l). Furthermore, most conventional αβ T cells do not develop in these mice and thus effector Th17 cells were extremely rare in *Bcl11b*$^{\Delta iThy}$ mice (Fig. 3i). Rather, MR1-reactive T cells possessing T$_{FH}$ potential accumulated (Fig. 3i, l). It is possible that these cells may provide B cell help for autoantibody production in a non-cognate manner as recently reported[35].

It is interesting that canonical type 2 MAIT cells have not been clearly defined[36], given that type 2 lymphocytes are a well-established and conserved subset across the wide variety of T lymphocytes and innate lymphocyte lineages[37–39]. The identification of Trav1$^{neg}$Traj33$^{pos}$ non-canonical MR1-reactive T cells that produce IL-4 might represent MR1-reactive type 2 T cells, together with recent reports on IL-4-producing MAIT cells[4]. Still, we cannot exclude the possibility that the absence of Bcl11b results in the derepression of IL-4-related genes in these cells[40]. Detailed immune profile analysis of clonotype #**1** T cells present in WT mice would clarify this issue.

The observation of incomplete symptom suppression in *Bcl11b*$^{\Delta iThy}$ mice on a μMT- or *Il4*$^{-/-}$-background suggests that antibodies are not the sole effectors contributing to these disorders. As previously proposed, the cytotoxic effector molecules, granzymes, could be an additional candidate for causing direct tissue damage[41]. The immune profile of the second most frequent T cell clone (clonotype #**2**) expanding in *Bcl11b*$^{\Delta iThy}$ mice might support this assumption; these cells included a granzyme B$^+$ population whose development was also dependent on MR1 and microbiota (Supplementary Fig. 3i, j).

Although Bcl11b is required for early T cell development, Bcl11b-null mice are neonatal lethal due to its importance in neural development[13]. Thus, genetic modification of Bcl11b specifically in immature thymocytes within 'viable' mice allowed us to detect previously unappreciated αβ T cell subsets. Such type 2 skewing of the remaining T cells may partly explain the exacerbated type 2 immunity in humans bearing hypomorphic Bcl11b mutations[42]. Currently, whether αβ T cells in *Bcl11b*$^{\Delta iThy}$ mice develop through the DP stage is unclear; it is possible that they develop into mature T cells from the

DN2 stage, in a process similar to an IFNγ-producing γδ T subset[10,43]. The reported "DN2 arrest phenotype of Bcl11b-deficient cells[6–8]" and the "requirement of pre-TCR-mediated β-selection (generally thought to accompany the DN to DP transition)" in the present study are apparently contradictory observations. Still, these T cells in *Bcl11b*$^{\Delta iThy}$ mice express CD5, a marker indicative of having experienced TCR engagement during positive selection. Typical MAIT and iNKT cells are positively selected by DP thymocytes and express PLZF, a lineage-defining transcription factor for these innate T lymphocytes[3]. However, a subpopulation of MR1-restricted T cells is selected by non-hematopoietic cells and is PLZF-negative[26], as observed in the T cells of *Bcl11b*$^{\Delta iThy}$ mice (Supplementary Fig. 2e). Given that antigen-presenting DP cells are lacking in the *Bcl11b*$^{\Delta iThy}$ thymus, T cells might be selected by thymic epithelial cells. The introduction of fine fate-mapping alleles in mixed chimera experiments may clarify this issue.

Under GF conditions, inflammatory symptoms in many autoimmune models worsen[44–46]. In contrast, the complete remission of inflammation in GF *Bcl11b*$^{\Delta iThy}$ mice indicates that the effector mechanisms are distinct from typical autoimmune diseases. Indeed, the generation of pathogenic T cells and inflammation in the current model require microbiota, presumably because microbiota provide antigens for MR1-restricted T cells, as was also recently shown for MAIT cells[2,26,41,47]. Similarly, as yet unidentified "mobile" metabolites from symbiotic bacteria may also activate the MR1-reactive T cells in the intestine and pancreas, which are anatomically connected via pancreatic ducts[48,49]. It is also possible that cytokine milieu, such as IL-12 or IL-1/18, present under SPF conditions might activate MR1-reactive T cells independently of the TCR[50].

A link between lymphopenia and autoimmunity has been reported[51]. Under lymphopenic conditions, the compensatory expansion of T cells exhibiting innate T cell/T$_{FH}$ cell profiles, converts these cells into pathogenic effectors[52–54]. Taken together, the present study suggests a possible contribution of non-canonical innate T cells to autoimmune diseases under T cell-imbalanced conditions and/or dysbiosis.

## Methods

### Mice
This study was approved by the Committee of Ethics on Animal Experiments in the Faculty of Medicine of Kyushu University, Science Research Center of Institute of Life Science and Medicine of Yamaguchi University and Research Institute for Microbial Diseases of Osaka University. Experiments were carried out under the control of the Guidelines for Animal Experiments. Mice were maintained under 12 h dark/light cycle and constant conditions of temperature (18–23 °C) and humidity (40–60%). C57BL/6 mice were purchased from SLC (Shizuoka, Japan). *Bcl11b*$^{flox/flox}$ mice were provided by R. Kominami (Niigata University, Japan). *Rag1*$^{Cre/+}$ mice were provided by T. Rabbitts (Leeds Institute of Molecular Medicine, United Kingdom) by courtesy of K. Akashi (Kyushu University, Japan). *Il4*$^{GFP/GFP}$ mice were provided by Dr. W.E. Paul (National Institute for Health, USA). μMT mice were provided by D. Kitamura (Tokyo University of Science, Japan). *Bcl11b*$^{flox/flox}$, *Rag1*$^{Cre/+}$ *Il4*$^{GFP/GFP}$. and μMT mice were backcrossed to

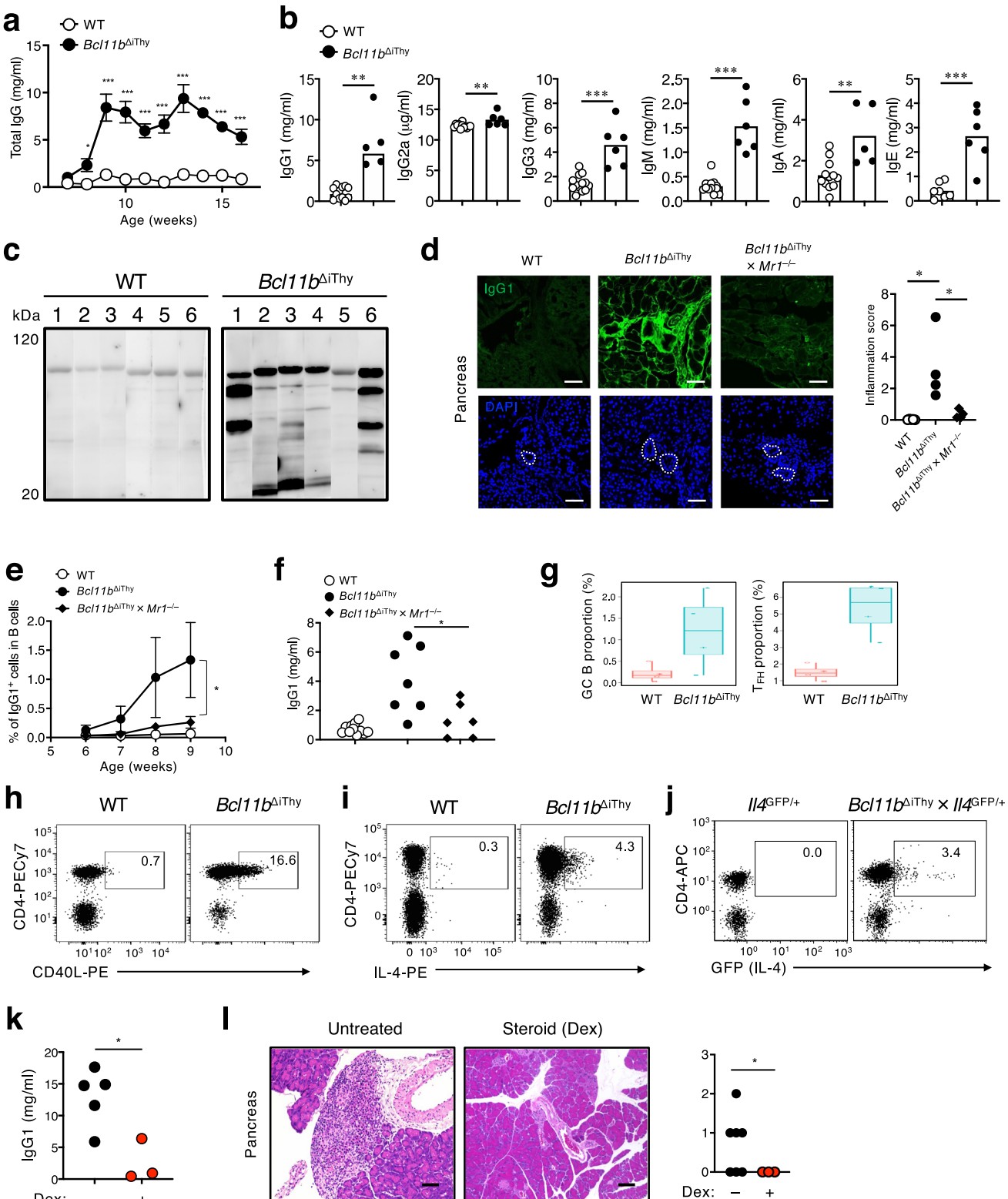

C57BL/6 mice more than ten generation. *Mr1*[−/−] mice provided by Dr. Susan Gilfillan (Washington University, USA) on the courtesy of Dr. K. Iwabuchi (Kitazato University, Japan) were backcrossed to C57BL/6 mice for at least twenty generations. Throughout the study, we mostly used littermate mice for control groups and designated Rag1-, Bcl11b- and MR1-sufficient mice (*Rag1*[Cre/+ or +/+] × *Bcl11b*[+/+] × *Mr1*[+/−], *Rag1*[Cre/+ or +/+] × *Bcl11b*[flox/+] × *Mr1*[+/−], *Rag1*[+/+] × *Bcl11b*[flox/flox or flox/+] × *Mr1*[+/+], *Rag1*[+/+] × *Bcl11b*[flox/flox] × *Mr1*[+/+ or +/−], *Rag1*[+/+] × *Bcl11b*[flox/+] × *Mr1*[+/+] mice) as WT mice. Germ-free *Bcl11b*[ΔiThy] mice are generated by in vitro fertilization at the

CLEA Japan. Germ-free mice were bred and maintained in vinyl isolators within the facility in CLEA Japan. For steroid treatment, *Bcl11b*[ΔiThy] mice were sequentially administrated from 5 weeks of age with an increased amount of dexamethasone (Cat. No.D4902, Sigma) (1, 3, 10 mg/kg) three times for every two weeks intraperitoneally, in accordance with human therapy[55] and analyzed at 12 weeks of age. For treatment with antibiotics, mice (5 weeks old) were given for 13 weeks with ampicillin (Cat. No.19769-64) (1 g/L), vancomycin (Cat. No.35575-94) (500 mg/L), neomycin sulfate (Cat. No.19767-42) (1 g/L), and

**Fig. 4 | Autoantibody production is dependent on MR1-reactive T cells.**
**a**, **b** Productions of total immunoglobulin ($n = 3$ biologically independent animals in each group) (**a**) and immunoglobulin subclass proteins at 12 weeks of age (**b**) in the sera of WT and $Bcl11b^{\Delta iThy}$ mice analyzed by ELISA. Asterisks indicate statistical significance determined by Two-way ANOVA (**a**) or by unpaired two-tailed Student's $t$-test (**b**) (*$p < 0.05$, **$p < 0.01$, ***$p < 0.001$). **c** Pancreatic homogenates from $Rag1^{-/-}$ mice were blotted with sera from six individual sick $Bcl11b^{\Delta iThy}$ mice (10–15 weeks old) in the lane-separated membrane. **d** IgG1 depositions determined by immunohistochemical analysis of sections of pancreas from the indicated strains of mice (15 weeks old). Dotted circles indicate pancreatic ducts. Scale bar: 100 μm. Quantified data of antibody deposited areas was shown on the right panel. *$p = 0.0121$ and 0.0387 by unpaired two-tailed Student's $t$-tests. **e** Graphs show the percentage of MR1-dependent IgG1$^+$ B cells ($n = 6$–10 biologically independent animals at each time point). **f** IgG1 production in 12-week-old indicated mice. *$p = 0.0149$ by two-way ANOVA (**e**) or *$p = 0.0234$ by unpaired two-tailed Student's

$t$-tests (**f**). **g** Percentages of germinal center (GC) B cells and $T_{FH}$ cells in WT and $Bcl11b^{\Delta iThy}$ mice (13–15 weeks old) analyzed by mass cytometry ($n = 4$ biologically independent animals). The center of the box is the median, bounds are the 75% and 25% percentiles. The whisker bounds are 1.5*the interquartile range past the 25% or 75% percentiles. All values are shown so the minima and maxima are just the top and bottom data points. **h**–**j** Surface CD40L expression (**h**), IL-4 production (**i**) and IL-4-GFP reporter fluorescence (**j**) in CD4$^+$ αβ T cells of WT, $Bcl11b^{\Delta iThy}$, $Il4^{GFP/+}$ and $Bcl11b^{\Delta iThy} \times Il4^{GFP/+}$ mice (10 weeks old). Numbers in dot plots show positive cells among total lymphocytes (**h**–**j**). **k**, **l** IgG1 productions (**k**) and histological analysis of pancreas by H&E staining (**l**) in $Bcl11b^{\Delta iThy}$ mice after steroid treatment with dexamethasone (Dex). **k**, **l** *$p = 0.0138$ (**k**), *$p = 0.0421$ (**l**) by unpaired two-tailed Student's $t$-tests. **l** Scale bar: 100 μm. **a**, **e** Data are expressed as mean ± SEM. **b**, **e**, **f**, **g** Data are combined from at least two independent experiments. Data are representative of two (**a**) and three (**h**–**j**) independent experiments.

---

metronidazole (Cat. No.23254-64) (1 g/L) in drinking water. All antibiotics were purchased from Nakarai Tesque, Inc.

### Cell lines
Reporter cell line expressing clonotype #**1** αβTCR or MAIT TCR was generated by retroviral gene transduction to TCR-negative mouse T cell hybridoma with an NFAT-GFP reporter gene, a kind gift from Dr. H. Arase (Osaka University, Japan). Mouse MR1-expressing antigen-presenting cells were generated by retroviral gene transduction of mouse MR1 to mouse fibroblastic cell line NIH3T3 (Cat. No.CRL1658, ATCC).

### Compounds
Ac-6-FP (Cat.No.11.418) was purchased from Schircks laboratories. 5-A-RU (Cat.No.A629245) was purchased from Toronto Research Chemicals. Methylglyoxal solution (Cat.No.M0252) and riboflavin (Cat.No.R9504) were purchased from Sigma-Aldrich. 5-OP-RU was generated by reacting 5-A-RU with equal molar ratio of methylglyoxal. 5-OP-RU concentrations are shown under assumption that all 5-A-RU is converted to 5-OP-RU.

### Flowcytometric analysis
αGC-loaded CD1d tetramers were prepared following the manufacturer's instructions (Cat No.TS-MCD-1, Medical & Biological Laboratories co., LTD). For staining of mouse MAIT cells, APC-labeled mouse MR1 tetramer provided by NIH Tetramer core facility was used[2]. FITC-conjugated anti-mouse (m) CD45.1 (A20, 1:40 dilution, Cat No.110705), anti-mCD45.2 (104, 1:100 dilution, Cat No.109805) mAbs, PE-conjugated anti-mCD5 (53-7.3, 1:100 dilution, Cat No.100607), anti-mCD8α (53-6.7, 1:80 dilution, Cat No.100707), anti-mCD40L/CD154 (MR1, 1:100 dilution, Cat No.106505), anti-mCD117 (2B8, 1:40 dilution, Cat No.105807), and anti-MR1 (26.5, 1:40 dilution, Cat No.361105) mAbs, PECy7-conjugated anti-mCD4 (RM4-5, 1:200 dilution, Cat No.100527) mAb, PerCP-Cy5.5-conjugated anti-mI-A/I-E (M5/114.15.2, 1:200 dilution, Cat No.107623) and anti-mCD44 (IM7, 1:200 dilution, Cat No.103031) mAbs, APC-conjugated anti-mTCRβ (H57-597, 1:40 dilution, Cat No.109211), anti-mCD4 (RM4-5, 1:100 dilution, Cat No.100515), anti-mIL-18Rα (A17071D, 1:100 dilution, Cat No.157905), anti-mCD25 (PC61, 1:200 dilution, Cat No.102011), anti-mCD44 (IM7, 1:200 dilution, Cat No.103011), anti-mCD45.2 (104, 1:40 dilution, Cat No.109813) and anti-mCD62L (MEL-14, 1:100 dilution, Cat No.104411) mAbs, AF700-conjugated mCD8 (53-6.7, 1:100 dilution, Cat No.100729) mAb, APCCy7-conjugated mTCRβ (H57-597, 1:40 dilution, Cat No.109219) mAb, BV421-conjugated mTCRβ (H57-597, 1:40 dilution, Cat No.109229) and mCD8α (53-6.7, 1:80 dilution, Cat No.100737) mAbs, BV510-conjugated mB220 (RA3-6B2, 1:100 dilution, Cat No.103247) and anti-mCD4 (RM4-5, 1:100 dilution, Cat No.100553) mAbs were purchased from BioLegend. FITC-conjugated anti-mIgG1 (A85-1, 1:50 dilution, Cat No.553443) mAb was purchased from BD Biosciences. PE-conjugated anti-mIL-18R (P3TUNYA, 1:80, Cat No.12-

5183-82) and APC-eFluor780 conjugated anti-mTCRβ (H57-597, 1:20, Cat No.47-5961-82) mAbs were purchased from Thermo Fisher Scientific. Dead cells were stained with 7AAD (7-amino-actinomycin D, Cat No.420403, BioLegend) and viable cells were analyzed by FACS Verse (BD Biosciences) or Gallios flow cytometer (Beckman Coulter). The data were analyzed using FlowJo software (TreeStar).

### Mass cytometry analysis
Antibodies were conjugated with the MaxPar conjugation kit according to the manufacturer's instructions with the exception of cisplatin-labeled antibodies. Metal isotopes were obtained from Standard Bio-Tools with the exception of Indium 113 and 115 (Trace Sciences). Conjugated antibodies were stored in PBS-based Ab stabilizer (Candor Biosciences) and all antibodies were titrated for optimal staining concentrations with control mouse tissues. Full details of the staining panel are provided in Supplementary Table 1. For staining, $1 \times 10^6$ cells per sample were initially barcoded with metal-conjugated CD45 antibodies in the presence of Fc blocker for 30 min at RT and then washed twice in CyFACS buffer (PBS with 0.1% BSA and 2 mM EDTA) and then barcoded cells were pooled. Pooled cells were stained with a metal-conjugated surface stain antibody cocktail for 45 min at RT. Cells were then washed twice in CyFACS buffer, stained for viability with the cisplatin analogue dichloro-(ethylenediamine) palladium (II) in PBS for 5 min at RT and then fixed and permeabilized using the Foxp3 Transcription Factor Staining Buffer Set according to the manufacturer's protocol (Thermo Fisher Scientific). Cells were subsequently stained with an intracellular antibody cocktail for 45 min at 4 °C. Cells were then washed twice in CyFACS buffer and once in PBS. Cells were then fixed overnight in 2% formaldehyde solution containing DNA Cell-ID Intercalator-103Rh (Standard BioTools). Prior to acquisition, cells were washed once in CyFACS buffer and twice in $H_2O$. Cells were then diluted to $1 \times 10^6$ cells/ml in $H_2O$ containing 15% EQ Four Element Calibration Beads (Standard BioTools) and filtered. Cells were acquired at a rate of 200–300 cells/s using a Helios mass cytometer (Standard BioTools). Flow Cytometry Standard files were normalized to EQ bead signal. Initial gating and debarcoding was performed in CytoBank software. Clustering was then performed after gating on B cells (B220$^+$CD3$^-$) or CD4 T cells (CD4$^+$CD3$^+$TCRβ$^+$B220$^-$) by FlowSOM within the CATALYST Bioconductor package referring to CyTOF workflow (version 3)[56]. Differential abundance analysis of clusters was performed with edgeR, calculating adjusted $p$-values.

### Bone marrow and T cell transplantation
BM cells from CD45.1$^+$ WT or CD45.2$^+$ $Bcl11b^{\Delta iThy}$ mice were suspended in phosphate-buffered saline (PBS) and $1 \times 10^7$ cells were injected into 8 Gy-irradiated $Cd3e^{-/-}$ mice. For mixed BM transfer, $Cd3e^{-/-}$ mice received 1:1 ratio of CD45.1$^+$ WT and CD45.2$^+$ $Bcl11b^{\Delta iThy}$ BM cells. The body weight and the frequency of CD4$^+$ T cells in the peripheral blood were analyzed weekly. For T cell transfer, $1 \times 10^7$ cells of CD4$^+$ T cells or

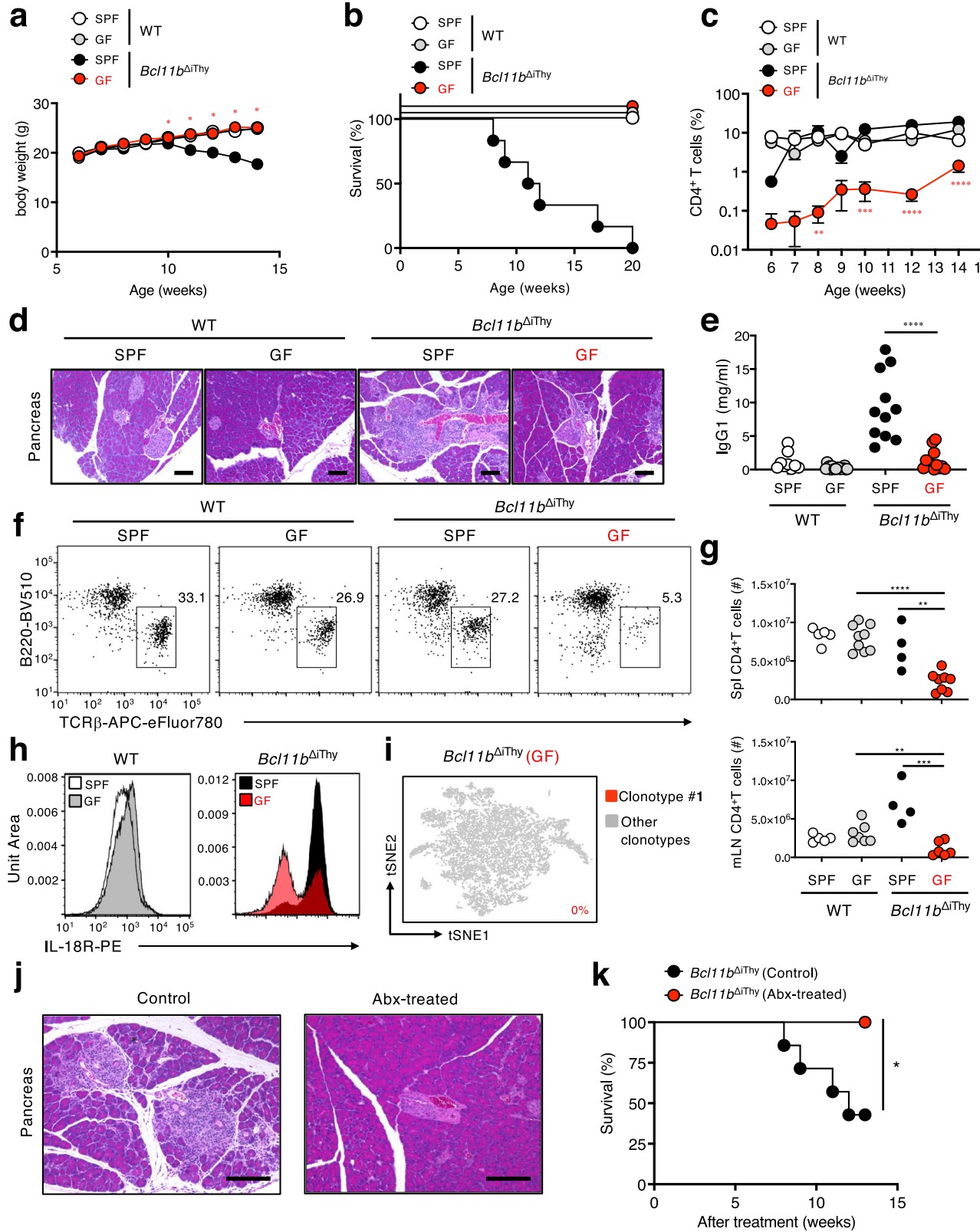

CD4$^+$ T cells lacking CD25$^+$ Treg cells from CD45.1$^+$ WT mice were transferred into CD45.2$^+$ *Bcl11b*$^{\Delta iThy}$ mice.

### Single cell-based transcriptome and TCR analysis
Libraries for mouse T cells were prepared using following reagents; Chromium Single Cell 5′ Library & Gel Bead Kit, PN-1000014; Chromium Single Cell A Chip Kit, PN-1000009; Chromium i7 Multiplex Kit,

PN-120262; Chromium Single Cell V(D)J Enrichment Kit, Mouse T Cell, PN-1000071. Approximately $2 \times 10^4$ cells are loaded into Chromium microfluidic chips to generate single-cell gel-bead emulsions using the Chromium controller (10X Genomics) according to the manufacturer's recommendations. Suspensions containing ~16,000 cells were loaded on the instrument. RNA from each sample was subsequently reverse-transcribed in a Veriti Thermal Cycler (Thermo Fisher Scientific), and

**Fig. 5 | Microbiota are essential for the development of the disorders in *Bcl11b*^ΔiThy mice. a–c** Age-related changes in body weight (**a**), survival rates (**b**) and CD4^+ T cell frequencies (**c**) for WT and *Bcl11b*^ΔiThy mice under SPF (WT mice (white), $n = 5$; *Bcl11b*^ΔiThy mice (black), $n = 8$) or GF conditions (WT mice (gray), $n = 18$; *Bcl11b*^ΔiThy mice (red), $n = 13$). **d** H&E staining of sections of pancreas of WT and *Bcl11b*^ΔiThy mice (18 weeks old) under SPF or GF conditions. Scale bar shows 100 μm. **e** Each plot represents amount of IgG1 present in the serum of the indicated mice (18 weeks old). **f** Frequencies of B cells (B220^+) and αβ T cells (TCRβ^+) within splenocytes from the indicated mice (16 weeks old) after gating on lymphocytes. **g** Each plot represents CD4^+ T cell number in the spleens (spl) and mesenteric lymph nodes (mLN) from the indicated mice (18 weeks old). **h** Histograms show IL-18R expression on αβ T cells from WT and *Bcl11b*^ΔiThy mice (18 weeks old) under SPF or GF conditions. **i** Single cell transcriptome and TCR analysis of sorted αβ T cells ($n = 7551$ cells) from *Bcl11b*^ΔiThy mice (18 weeks old) maintained under GF conditions.

Numbers in the tSNE plot show the frequency of clonotype #1. **j** H&E staining of sections of pancreas of mice (16 weeks old) after treatment with or without antibiotics (Abx). Scale bar shows 60 μm. **k** Graphs show survival of *Bcl11b*^ΔiThy mice after treatment with (red) or without (black) antibiotics (Abx) ($n = 9$ and $n = 6$ biologically independent animals respectively). Asterisks indicate statistical significance between GF and SPF conditions of *Bcl11b*^ΔiThy mice determined by two-way ANOVA test (**a, c**) (*$p < 0.05$, **$p < 0.01$, ***$p < 0.001$, ****$p < 0.0001$), unpaired two-tailed Student's *t*-test (****$p < 0.0001$ (**e**); ****$p < 0.0001$, **$p = 0.0033$ (spl), **$p = 0.0082$, ***$p = 0.0009$ (mLN) (**g**)), and logrank test ($p = 0.0339$) (**k**) respectively. **c** Data are expressed as mean ± SEM. Gating strategies are shown in Supplementary Fig. 6c for Fig. 5h, and Supplementary Fig. 6d for Fig. 5f. **d, f, h, j** Data are representative of three independent experiments. **a, b, c, e, g, k** Data are combined from at least two independent experiments.

all subsequent steps to generate single-cell libraries were performed according to the manufacturer's protocol, with 14 cycles used for cDNA amplification. Then ~50 ng of cDNA was used for gene expression library amplification by 14 cycles in parallel with cDNA enrichment and library construction for T cell libraries. Fragment size of the libraries were confirmed with the Agilent 2100 Bioanalyzer (Agilent). Libraries were sequenced on an Illumina NovaSeq 6000 as paired-end mode (read1: 28 bp; read2: 91 bp). The raw reads were processed by cellranger 3.1.0 (10X Genomics). Clonotype analysis was done using Loupe Cell Browsers provided by 10x Genomics (https://support.10xgenomics.com/single-cell-gene-expression/software/downloads/latest#loupetab). After identification of TCR clonotypes by the loupe V(D)J browser, transcriptome analysis of these clusters was further performed by the loupe browser.

### Bulk RNA-sequencing analysis
Single-cell suspensions from splenocytes were dissolved in the TRIzol (Cat. No.15596026, Thermo Fisher Scientific) and applied for bulk RNA-sequencing analysis was applied for TCR repertoire analysis using the SMARTer Mouse TCR a/b Profiling Kit (Takara Bio USA Inc).

### Pseudotime analysis
For inferring cell lineages and pseudotimes, a Bioconductor package "slingshot[57]" and their lineage inference workflow to predict cell lineage (s) and pseudotime (s) were used. The output value "filtered_feature_bc_matrix" from the Cell Ranger was used as count data for slingshot. An R clustering package "mclust" was applied to the t-SNE matrix generated by Cell Ranger to infer clusters, yielding a final repertoire of 7 clusters ("chartreuse", "cyan", "khaki", "plum", "orange", "orangered", "sienna"). The "getLineages" function of slingshot was used to obtain lineage structure of the t-SNE matrix and their pseudotime (s). We specified the initial cluster as chartreuse based on gene expression of Mki67. We also used the "getCurves" function of slingshot to construct smooth curve in each lineage which inferred by getLineages function using default parameter setting except for extend = "n" and stretch = 0. For gene expression in the t-SNE dimension, the natural logarithm of the given value plus one (log1p) of the count data of each gene was plotted by plotGeneCount functions of a Bioconductor package "tradeSeq[58]". We customized the plotGeneCount script to change color and position of the log1p color scales (Supplementary_code.txt).

### Preparation of fractionated tissue homogenates
After snap-frozen tissues in liquid nitrogen were smashed in a mortar, homogenates were centrifuged ($15,000 \times g$ for 20 min at 4 °C). The homogenates were resuspended in water with 0.1% TFA and fractionated by HPLC (JASCO LC-NetII/ADC) using COSMOSIL 5C$_{18}$-MS-II 4.6 mmID × 250 mm (Cat. No. 38020-41, Nakalai tesque). After loading, 70 fractions were collected by increasing Acetonitrile to 80%. Fractions were dried and stored at −80 °C until use.

### Reporter assay
Reporter cell lines were cocultured with mouse MR1-expressing NIH3T3 cells in the presence of homogenates[59]. One day after the coculture, GFP expressions in reporter cells were analyzed by flowcytometry after gating on CD3^+ reporter cells.

### Histological analysis
Histological analysis was performed after freshly isolated tissue samples were fixed with formalin and then embedded in paraffin or OCT compound (Cat No.45833, Sakura Finetek). Sections were stained with Hematoxylin and Eosin (H&E) for detecting inflammatory cells or with Sirius red for the assessment of collagen contents using Picrosirius Red Stain Kit (Cat. No.24901, Polyscience). OCT compound-embedded sections of mouse pancreatic tissues were stained with antibodies against mIgG1 (0.5 μg/ml, A85-1, Cat. No.553443, BD Biosciences), mCD4 (10 μg/ml, GK1.5, Cat. No.100407, BioLegend) or B220 (5 μg/ml, RA3-6B2, Cat. No.103225, BioLegend) or biotinylated PNA (20 μg/ml, Cat. No.BK-1000, Vector laboratories) overnight at 4 °C. For PNA staining, Alexa Fluor 647-conjugated Streptavidin (1 μg/ml, Cat. No.405237, BioLegend) was applied on the tissues and incubated for 2 h at 37 °C. Stained tissues were visualized using an FV3000 confocal microscope (Olympus). Inflammatory score was determined as described previously[60]. The degrees of cell infiltration judged by histology using H&E staining of pancreatic tissues were scored as follows: 0, no detectable infiltration; 1, a focus of perivascular infiltration; 2, several foci of perivascular infiltration; 3, cellular infiltration in more than 50% of vasculature; 4, cellular infiltration in more than 80% of vasculature; and 5, massive infiltration. For quantification of the percentage of Picrosirius Red-positive area, images of pancreatic tissue were binarized using a fixed threshold by ImageJ and positive pixel ratio was calculated.

### Determination of isotype-specific immunoglobulins, blood glucose levels, and lipase activities in serum
Concentrations of polyclonal mIgM (Cat. No. E99-101), mIgG1 (Cat. No. E99-105), IgG2a (Cat. No. E99-107), mIgG3 (Cat. No. E99-111) and mIgA (Cat. No. E99-103) were analyzed by isotype-specific enzyme-linked immunosorbent assay (Bethyl Laboratories, Inc). Blood glucose level was analyzed by ACCU−CHEK Aviva System Blood Glucose Meter (Roche). Lipase activity was analyzed with Lipase Color Liquid (Roche).

### Immunoblot analysis
For analysis of sorted CD4^+ αβ T cells, cells were lysed for 5 min at 95 °C in SDS lysis buffer (20 mM EDTA, 10 mM Tris HCl, 150 mM NaCl and 1% NP40). For analysis of pancreatic homogenates, tissues from Rag1-deficient mice were sonicated and then suspended in RIPA buffer. Proteins were resolved by SDS-PAGE and transferred to Immobilon−P membranes. Membranes were blocked by Blocking One (Cat. No. 03953-66, Nakalai tesque) and were detected by immunoblot analysis. The rabbit anti-mBcl11b polyclonal Ab was kind gift from Yukio

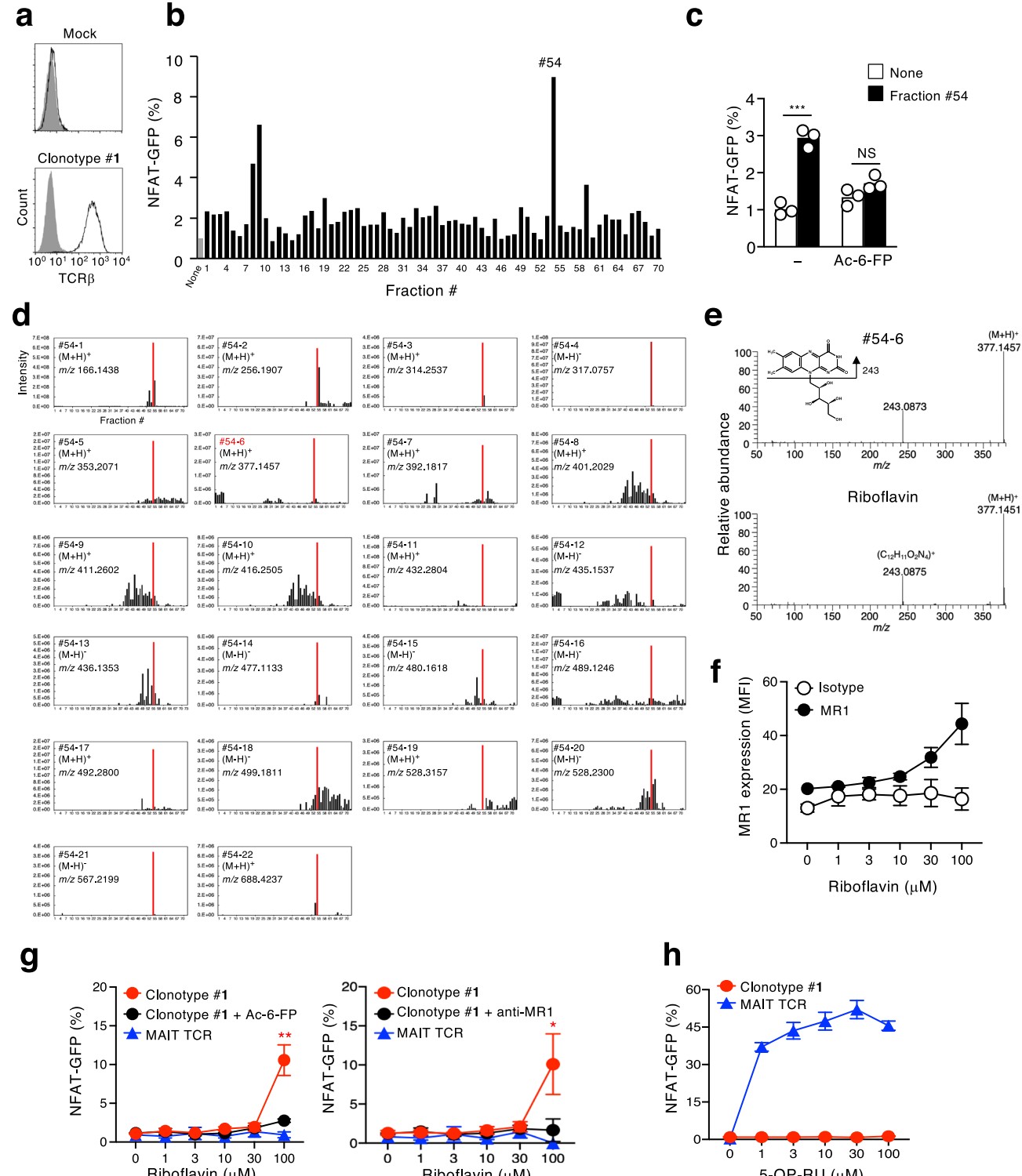

**Fig. 6 | Clonotype #1 recognizes riboflavin in an MR1-dependent manner.**
**a** Histograms show clonotype #1 TCR expression as compared with the parental cell line (Mock). These cells were stained with isotype control (closed) and anti-TCRβ antibody (open). **b, c** Graph shows NFAT-GFP expressions in the reporter cell line expressing clonotype #1 TCR after cocultured with fractionated fecal homogenates in the presence of mouse MR1-expressing NIH3T3 cells. **c** Ac-6-FP (10 μM) was used to block interactions between MR1 and TCR. ***$p = 0.0004$ by unpaired two-tailed Student's $t$-test. NS indicates not significantly different. **d** Plots of the ion intensity of precursor ions contained in the fraction of #54. #54-6, having similar $m/z$ value to exact mass of known metabolites in the human metabolome database, was highlighted in red. **e** HRMS/MS spectra of riboflavin; fraction #54-6 (upper) and authentic standard (lower) in the positive ion mode. **f** MR1 expressions on MR1-expressing thymoma 15 h after co-culture in the presence of riboflavin were analyzed using a PE-labeled anti-MR1 antibody. MFI indicates mean fluorescence intensity of the MR1 expression. **g, h** GFP-reporter activities of cells expressing the clonotype #1 TCR or MAIT TCR after stimulation with riboflavin (**g**) or 5-OP-RU (**h**) in the presence of mouse MR1-expressing NIH3T3 cells with or without Ac-6-FP (100 μM) or anti-MR1 antibody (10 μg/ml). 5-OP-RU concentrations are indicated as described in the Methods section. **g** Asterisks indicate statistical significance determined by unpaired two-tailed Student's $t$-test (**$p = 0.0024$, *$p = 0.0241$). **c, f, g, h** Data are expressed as mean ± SEM. Gating strategies are shown in Supplementary Fig. 6e for Fig. 6a. **a, c, f, g, h** Data are representative of two independent experiments.

Mishima (Niigata University). Anti-β-actin (2F1-1) mAb (Cat. No.622101, BioLegend) was used as an internal control. For pancreatic homogenates, serums from indicated mice were used as the first antibody. The first antibody was detected by peroxidase-labeled Goat anti-mIgG (H + L) (Cat. No.115-035-003, Jackson ImmunoResearch Laboratories Inc.). Signals were detected using LAS4000mini (GE healthcare).

## Electrospray ionization-quadrupole Orbitrap high-resolution tandem mass spectrometry (ESI-HRMS/MS) analysis and data mining

Soluble fractions were dissolved in 50 µl of methanol/water (95:5, vol/vol). Flow injection analysis was performed using a Nexera LC system (Shimadzu Co., Kyoto, Japan) coupled with Q Exactive, a high performance bentchtop quadrupole Orbitrap mass spectrometer (Thermo Fisher Scientific, Waltham, MA), equipped with an electron spray ionization source. The flow injection conditions in the positive and negative ionization modes were described previously[61]. Data mining procedure was described previously[27]. In short, Compound Discoverer 3.0 software (Thermo Fisher Scientific) was used for data processing, involving peak detection, data grouping, and filling gaps. Differential analysis (fraction #54 vs. other fractions, fold change > 5) was performed using an XLSM extension developed with Visual Basic for applications in Microsoft Excel 2013. Among more than 20,000 ion peaks (including isotopes and adducts detected in the positive and negative ionization), 22 candidates were detected selectively and reproducibly present in fraction #54.

## Expression and purification of clonotype #1 ectodomain

Expression and purification of clonotype #1 αβ ectodomain was performed as described recently[62]. In brief, both clonotype #1 α and β ectodomains were expressed as fusion proteins with N-terminal hexahistidine tag and Tabaco Etch Virus (TEV) protease cleavage site. The clonotype #1 α and β ectodomains were separately expressed in *Escherichia coli* as inclusion bodies. The inclusion bodies were solubilized with 6 M guanidine hydroxychloride. The solubilized clonotype #1 α and β ectodomains were mixed, refolded, and then dialyzed. Clonotype #1 αβ heterodimer was applied to Ni-NTA agarose. After the removal of histidine tag by TEV protease treatment, the proteins were applied to size exclusion and anion change chromatography. The purities of proteins were assessed by SDS-PAGE.

## Crystallization, data collection, and structure determination of clone #1 ectodomain

All crystallization trials were performed by sitting drop vapor diffusion method as described previously[62]. Initial crystallization conditions were searched using Index (Hampton Research) and SG1 Screen (Molecular Dimensions). Diffraction-quality crystals were obtained under the condition of 0.1 M Bis-tris (pH 5.5), 0.2 M lithium acetate and 17% (w/v) polyethylene glycol 10,000 at 293 K. Crystals were soaked into reservoir containing 10% (v/v) glycerol and rapidly frozen in liquid nitrogen. X-ray diffraction data were collected on the beamline BL-1A at Photon Factory (Tsukuba, Japan). All data sets were integrated with program XDS and scaled with the program Scala. The phases of the crystal were determined by molecular replacement with program Molrep using TCR-017 ectodomain (PDB code: 7EA6) as a search model. Further model building was manually performed with program Coot. Refinement was initially conducted using program Refmac5 and then Phenix.refine for final model. The stereochemical quality of the final model was assessed by Molprobity. Data collection and refinement statistics were summarized in Supplementary Table 2. Structural factors and coordinates have been deposited on Protein Data Bank under accession code 7F5K. All structural figures were depicted by program PyMOL2.0 (The PyMOL Molecular Graphics System, Version 2.0 Schrödinger, LLC.). Structural superposition was performed with program SUPERPOSE.

## Statistical analyses

Statistical significance was calculated using Prism software (GraphPad, San Diego, CA). Differences with values for $p < 0.05$ were considered statistically significant.

## Reporting summary

Further information on research design is available in the Nature Portfolio Reporting Summary linked to this article.

## Data availability

Bulk RNA-sequencing and single-cell RNA-sequencing data have been deposited at the DNA Data Bank of Japan (DDBJ) database under the accession number DRA011320 within the Bioproject PRJDB10983. Structure factor and 3D coordinates of clone #1 TCR ectodomain have been deposited in the Protein Data Bank under the accession code, 7F5K. Source data are provided with this paper.

## Code availability

The customized plotGeneCount script and source data for Fig. 3k are provided as Supplementary Code. The script code provided in the supplementary material is available under the MIT License (Copyright 2022, Takahide Hayano). The original code was included in the package "tradeSeq" (https://bioconductor.org/packages/release/bioc/html/tradeSeq.html) which is distributed under the MIT License (Copyright 2019, Koen Van den Berge; Hector Roux de Bezieux). See https://opensource.org/licenses/mit-license.php for the full text of the MIT license.

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

## Acknowledgements

We thank S. Iwai, M. Kurata, S. Mondoon, M. Kawano, M. Shiokawa, K. Toyonaga, A. Kubota, X. Lu, Y. Harima, A. Tanaka, E. Ito, T. Oono, K. Kawabe, D. N. Azizah and T. Ito for technical support; S. Inuki, KH. Sonoda and Y. Yoshikai for discussions; M. Tanaka and K. Kaseda for embryonic engineering; the Cooperative Research Project Program of the Medical Institute of Bioregulation, Kyushu University and Science Research Center of Yamaguchi University for technical support. We also thank the beamline staff at the Photon Factory for crystallographic data collection. This research was supported by AMED (JP20gm0910010, JP20ak0101070, and JP20fk0108075), JSPS KAKENHI (JP20H00505 and JP16K08740), the Kurozumi Medical Foundation (K.S.), and the Takeda Science foundation (M.F.S.). The MR1 tetramer technology was developed jointly by Dr. James McCluskey, Dr. Jamie Rossjohn, and Dr. David Fairlie, and the material was produced by the NIH Tetramer Core Facility as permitted to be distributed by the University of Melbourne.

## Author contributions

K.S. and S.Y. conceptualized research; K.S., C.M., T.S., E.I., Y.I., M.T., and M.N. did investigation; N.F. and Y.O. provided resources; K.S., C.M., D.M., D.O., J.B.W., T.H., Y.A., and T.B. did data curation; M.F.S., M.S., and S.Y. supervised the research; K.S. and S.Y. wrote the manuscript.

## Competing interests

The authors declare no competing interests.

## Additional information

[1]Department of Microbiology and Immunology, Graduate School of Medicine, Yamaguchi University, Ube 755-8505, Japan. [2]Department of Ophthalmology, Department of Ocular Pathology and Imaging Science, Graduate School of Medical Sciences, Kyushu University, Fukuoka 812-8582, Japan. [3]Department of Molecular Immunology, Research Institute for Microbial Diseases, Osaka University, Suita 565-0871, Japan. [4]Division of Infection and Immunity, Joint Research Center for Human Retrovirus Infection, Kumamoto University, Kumamoto 860-0871, Japan. [5]Laboratory of Molecular Immunology, Immunology Frontier Research Center, Osaka University, Suita 565-0871, Japan. [6]Department of Infection Metagenomics, Genome Information Research Center, Research Institute for Microbial Diseases, Osaka University, Suita 565-0871, Japan. [7]Single Cell Genomics, Human Immunology, World Premier International Research Center Initiative Immunology Frontier Research Center, Osaka University, Suita 565-0871, Japan. [8]Genome Information Research Center, Research Institute for Microbial Diseases, Osaka University, Suita 565-0871, Japan. [9]Division of Metabolomics, Medical Institute of Bioregulation, Kyushu University, Fukuoka 812-8582, Japan. [10]Department of Medicine and Bioregulatory Science, Graduate School of Medical Sciences, Kyushu University, Fukuoka 812-8582, Japan. [11]Laboratory of Human Immunology (Single Cell Immunology), World Premier International Immunology Frontier Research Center, Osaka University, Suita 565-0871, Japan. [12]Laboratory of Experimental Immunology, World Premier International Immunology Frontier Research Center, Osaka University, Suita 565-0871, Japan. [13]Department of Systems Bioinformatics, Graduate School of Medicine, Yamaguchi University, Ube 755-8505, Japan. [14]Japan Agency for Medical Research and Development, Core Research for Evolutional Science and Technology, Tokyo 100-0004, Japan. [15]Department of Molecular Medicine and Metabolism, Research Institute of Environmental Medicine, Nagoya University, Nagoya 464-8601, Japan. [16]Systems Biochemistry in Pathology and Regeneration, Graduate School of Medicine, Yamaguchi University, Ube 755-8505, Japan. [17]Division of Molecular Design, Medical Institute of Bioregulation, Kyushu University, Fukuoka 812-8582, Japan. [18]Division of Molecular Immunology, Medical Mycology Research Center, Chiba University, Chiba 260-8673, Japan. ✉e-mail: yamasaki@biken.osaka-u.ac.jp

