## [Peer Review File · Nature Communications]

Symbiotic bacteria-dependent expansion of MR1-reactive T cells causes autoimmunity in the absence of Bcl11bREVIEWER COMMENTS

Reviewer #1 (Remarks to the Author):

Summary

In addition to MR1-5-OP-RU reactive MAIT cells, an increasing number of reports are emerging of other MR1-reactive T cells. However, their protective or pathogenic immune roles are unknown. Here, the authors describe the development of pancreatitis in mice where Bcl11b is ablated in immature thymocytes. They show the accumulation of pathogenic CD4⁺ MR1-reactive T cells in these mice, which appear to drive the disease progression. Based on reactivity to fractionated tissue homogenates, followed by mass spec analysis the authors identify riboflavin as an antigen presented by MR1 and recognised by the aberrant T cells.

This paper describes potentially exciting findings and contains several aspects that will be of interest to the field. However, there are many unanswered questions that would need to be addressed prior to publication. In particular, the later parts of the story around antigen identification would need significant further work to confirm what could be very interesting and important findings. Specific comments are provided below.

Major comments

1) The first few figures show an interesting phenotype in the Bcl11b deficient mice. One issue is the small numbers of mice presented for several of the figures (e.g. parts of figures 1, 2, 6 show only n=3-5 mice). The authors should ensure that these are representative of several repeats using age-matched mice. Fig 2B (showing no CD1d-aGC or MR1-5-OP-RU reactive cells in the spleens shows only one mouse for each).

2) The role of MR1 in thymic selection vs peripheral reactivity is topical in the MAIT/MR1T cell field. It is possible that MR1T cells (other than MAIT cells) may be selected on MHC molecules and cross-reactive on MR1. However, this study shows that the aberrant T cells did not develop when crossed to MR1^{-/-} mice. I feel the authors could build on this part of the story, which would be of great interest to the field.

3) The aberrant T cells don't accumulate when mice are crossed with MR1^{-/-}. The authors state they are not MAIT cells based on lack of PLZF expression. A recent publication (Drashansky et al. *iScience* 24(4): 102307, April 2021) showed that MAIT cell development is impaired in Bcl11b deficient mice, particularly MAIT-17 cells due to the blockade of PLZF and ROR γ t, needed for their development. This study should be considered and referenced.

4) The authors use the lack of TRAV1, lack of PLZF and lack of reactivity of one clone to MR1-5-OP-RU as evidence that these cells are not MAIT cells. Although not essential, it would be interesting to understand the location of these cells within various organs – do they have a mucosal distribution? What is their chemokine receptor expression?

5) There are many questions around the later part of the study looking at the identification of the antigen.

i) The authors show a structure of a TCR superimposed on MR1-6-FP, but seem to state later that this TCR is reactive towards MR1 presenting riboflavin. Does MR1 refold with riboflavin?. Alternatively can MR1-riboflavin modelled with the TCR in a similar way to the MR1-6-FP structure? If riboflavin sits differently in the cleft the TCR docking with MR1-6-FP is not informative here.

ii) Although clonotype 1 was most frequent, many other clonotypes were seen (Fig 3A). Were these also reduced on an MR1^{-/-} background? Were these reactive with MR1-riboflavin? Key results supporting riboflavin as an antigen in Fig 7F, G need further validation. Is the upregulation of surface MR1 specific (an MHC I control could be used). What do the error bars represent in these figures?

iii) from an evolutionary perspective, detection of riboflavin precursor compounds (ie 5-OP-RU) makes

sense, whereas the detection of riboflavin may lead to autoimmunity. The authors should speculate on the physiological relevance of this in normal mice, particularly given the response (Fig 7G) is much less sensitive than that of MAIT TCR for 5-OP-RU.

6) Has the reduction in MAIT cells in AIP patients been reported previously? The authors note that MAIT cells are reduced and MR1T cells are present at normal levels in these patients. They then remark that the ratio of these cells is altered (Fig. 5C), which seems obvious. Numbers rather than ratio should be shown for MR1-Ac-6-FP reactive cells from multiple donors.

7) In both mice and humans the authors refer to the balance of MR1T cells compared to MAIT cells or other T cells being important for pathology. However, to my knowledge no regulatory function has been attributed to MAIT cells. Do the authors consider the presence of MAIT cells decreases the ability of the pathogenic cells to expand (niche effect)? This could perhaps be addressed in the mice by the transfer of titrated numbers of cells.

8) Was any attempt made to identify the other m/z compounds? It seems unlikely that the one known compound would be identified as the antigen. It is possible that other related compounds are more potent antigens or MR1 ligands. In this case the techniques used to identify 5-OP-RU (refolding MR1 in the presence of the positive fraction) could be used for ligand identification.

9) It is unclear from the data shown whether the clonotype 1 TCR recognises MR1-riboflavin or MR1 independent of antigen (as has previously been described for other MR1-reactive T cells). In this case riboflavin may not be an antigen but may cause reactivity by non-specifically upregulating MR1. Does the clone also respond to MR1 in the presence of Ac-6-FP? The ternary structure as well as further functional analysis would be important here in understanding the specificity. In the case that this clone is not recognising a specific antigen, how does this affect the interpretation of the TCR extending into the Ag-binding cleft as modelled for MR1-6-FP?

Minor comments

1) The naming of MR1-reactive T cell subsets is still unclear in the field. The authors should consider their use of "MR1-restricted" vs "MR1-reactive" since it is possible that some MR1-reactive T cells are cross-reactive.

2) Change of tense in abstract and in first section of results.

3) Fig. 1.

i) B says n=5, but it seems only 4 mice are shown.

ii) Naming/positioning of panels D-F should be reconsidered (also for other figures).

iii) What age are the mice shown in C-E?

iv) In F, further explanation of the glucose levels should be added to the text or legend. The authors state that mice did not develop diabetes, but do show a statistical difference (albeit in only 3-5 mice)

4) Supp fig 1B. histology comparison of WT and Bcl11^{biThy} mice should be shown on the same scale.

5) 5-OP-RU is not listed in the methods, but is presumably generated from 5-A-RU and methylglyoxal, which are listed. The method for this and the validation of concentration should be shown. Previous studies have shown this reaction to be inefficient, so if the stated concentrations for 5-OP-RU on Figure 7H are really indicating 5-A-RU used in the reaction this should be stated, since this is likely an overestimate of 5-OP-RU concentration.

6) The methods state that the human MR1-K43A tetramer was used. Why was this chosen over the WT tetramer (or is this an error - since the NIH facility is listed as a source for mouse tetramers)? No source is noted for the K43A tetramer. Was this produced in-house? How does this impact the interpretation since the binding will differ between MR1-WT and MR1-K43A?

7) Figure 7A. The text (page 17) states that fecal extracts were fractionated and tested for their stimulatory capacity. However the figure legend refers to fractionated tissue homogenates. Which is correct?

Reviewer #2 (Remarks to the Author):

Shibata et al in their manuscript "Symbiotic dependent expansion of MR1-reactive T cells causes pancreatitis in the absence of BCL11b in lymphocytes." Utilizes many techniques to establish the cause of pancreatitis in thymic specific BCL11b knock out mice. The authors beautifully show that there is an expansion of pathogenic non-MAIT MR-1+ abT cells that induce B cells and pancreatitis in a microbiota dependent manner. Moreover, the authors also evaluate patients with autoimmune pancreatitis to find some similarities with their model. These are potentially exciting findings, but some of the connections and rigor of the experiments are lacking.

Major points:

1. One of the key findings of the manuscript is identification of a novel Mr1+ non MAIT abT cell population. However, very little is done to evaluate this population beyond the prevalence of clone 1 in the population. The authors used advanced techniques such as scRNAseq and CyTOF and yet the analysis of these techniques is very rudimentary, lacking in deep analysis of the data. 33 markers are used in the CyTOF panel, yet there is no comment on what the T cell or B cell population are in the BCL11b k/o mice with the main figure only commenting on the GC B cell and Tfh cell proportion proportions in the ko mice. What are the markers of the T cells and B cell in these k/o mice? Are there unique markers that identify the novel Mr1 population? What are the overall proportion of the other immune cells in these mice. Similarly, the scRNAseq data is not developed. What are the various clusters found, what is driving these clusters. The tsne plots for the single k/o and Mr1 ko are generated separately. It would be helpful to combine into one analysis. With common cluster identification, etc. Where wt mice analyzed? Was clone1 present in wt? What is the shift in the T cell populations between the wt and single ko mice, how is this altered by additional Mr1 ko? The scRNAseq data suggests that there is increase in Tfh cells in ko mice, are these markers present in the CyTOF data? or validated in any other way? Flow? Etc.
2. All antibody subclasses were drastically increased in the periphery not just IgG. Are these clonally expanded? Are they contributing to the disease? B cell combined ko has significantly increased survival arguing that it is the B cells not the T cells themselves that are driving the pathology. Can B cells from these mice be transferred to cause disease?
3. Do the authors believe that the mice are dying from pancreatitis? Why did the authors choose to examine only the pancreas and the large intestine.
4. In the BCL11b k/o + wt transfer experiments, the authors conclude that the restoration of the normal cells precludes expansion of k/o cells. It's possible that the wt cells have a competitive advantage, right? And just outcompete the k/o cells. There are almost no T cells from the k/o mice in the mixed chimera model. Unlike the B cells that are still present. Could it be that these are more prone to apoptosis?
5. Fig1I the percent of abT cells in peripheral blood of ko mice is almost 40% of the PBMCs. Is this due to an overall increase in these cells or proportional decrease in other immune cell populations. What is happening to the other immune populations?
6. Another key finding of the manuscript is identification of riboflavin as the metabolite driving these Mr1+ T cells, can the authors show that riboflavin is sufficient to drive pathology in the GF mouse experiments?

Minor points:

1. Methods need to be much more developed so that work can be reproduced. For example, Fig 1E is inflammatory score- how is this developed, what is counted. Similarly, all through out, there is lack of attention to details (on what day were the blood glucose collected, or the lipase? When did the steroid treatment start? Etc).
2. There is a lack of rigor to the histological and IF data. Most histological images are not quantified. And only one, presumably best example, is shown. For example, Fig1D increase in Sirius red stain

- between wt and ko, please quantify. Fig2 E and Fig2K, quantify immunologic score. Figure S1B, quantify thickness. S1C show the colons, etc.
3. Stylistically, please reformat the figures and the text so that both appear in chronological order (so that the sub-figures appear in the figures and within the text in chronological order, a before b, etc). It's challenging to follow in the current state and distracts from the manuscript.
 4. Nature communication is intended for a general audience, please label portions of the pancreas that the text is referring to
 5. T cell specific abstract misses the other important findings of the manuscript namely B cell and microbiota contribution to the disease
 6. Page 6 statement that T cells are required but not B cells is misleading as B cells clearly contribute to the phenotype from the data presented in the manuscript.
 7. It looks like TCRD double ko mice have a worsened phenotype than the BCL11b k/o, are they perhaps ameliorating the phenotype?
 8. IgG1 deposit pictures in Fig 4D are of poor quality and need to be quantified.
 9. "The inflamed pancreases of Bcl11b Δ iThy mice were infiltrated with IgG1+ B cells and CD4+ $\alpha\beta$ T cells." However, the staining is for B220, how do we know these are IgG+?
 10. Steroid decreased the levels of IgG, what about other antibodies? Was the increase in steroids started at birth? After pathology developed?
 11. In regards to the scRNAseq analysis of human data. The authors state that the naïve clusters are not well-defined in patients, this is very subjective. Was this data automatically clustered? Are there percent differences between the naïve clusters? What are the clusters? What are their markers? Similarly, authors state that green Tf cluster is well apparent in the disease subjects. This is again very subjective, as it looks like it is only apparent in case #10 with no detectable population in #9 and a very small population in #11. Similarly CD161 cluster is apparent in case #9 and 11. What clusters express CD69? How is the figure 5C data generated?

Reviewer #3 (Remarks to the Author):

Shibata K et al demonstrate that mice lacking a T cell-specific transcription factor, Bcl11b in immature thymocytes (Bcl11b Δ iThy mice) develop chronic pancreatitis. Although the pancreatitis was ameliorated by MR1 deficiency or under germ-free condition, MR1-restricted mucosal-associated invariant T(MAIT) cells were not increased in Bcl11b Δ iThy mice. Instead, they found that MR1-restricted TRAJ33-bearing T cells were expanded in these mice. They demonstrated that adoptive transfer of bone marrow from Bcl11b Δ iThy mice into CD3e $^{-/-}$ induced pancreatitis, but co-transfer of bone marrow from Bcl11b Δ iThy and WT mice failed to cause pancreatitis, suggesting that MR1-restricted T cells which were increased by the homeostatic expansion of these cells appear to play roles in the development of pancreatitis. They also showed that activated MR1-restricted T cells are present in the peripheral blood of patients with autoimmune pancreatitis. The pancreatitis in Bcl11b Δ iThy mice was accompanied with elevated autoantibodies against lysates of pancreas, expansion of germinal center B cells and T follicular helper cells. They characterize MR1-restricted TRAJ33-bearing T cells from Bcl11b Δ iThy mice by RNA sequencing and the crystal structure of Trbv7-6-Traj33/Trbv13-2-Trbj1-1 TCR. These MR1-restricted T cells recognized riboflavin presented by MR1 and displayed type 2 T and Tfh cell phenotypes. While this study is interesting, there are major concerns about the data shown.

Major Concerns:

Based on experiments with MR1 deficient mice, MR1-restricted T cells clearly contribute to the pathogenesis of Bcl11b Δ iThy mice. However, major questions concerning the function of MR1-restricted TRAJ33-bearing T cells remain open because they studied this T cell population under extremely unusual condition where T cell development is enormously impaired. In addition, the exact mechanism by which MR1-restricted TRAJ33-bearing T cells contribute to the pancreatitis is unknown. Do these MR1-restricted accumulate in the pancreas and cause inflammation, or activate autoreactive B cells? Would CD3e $^{-/-}$ mice still develop pancreatitis when transferred with restricted TRAJ33-bearing T cells? If restricted TRAJ33-bearing T cells are activated by recognizing riboflavin

generated by symbolic microbes, why is the main disease in Bcl11b Δ iThy mice is pancreatitis, not colitis? The generation of autoantibodies against pancreatic lysates was accompanied by enhanced germinal center B cell and Tfh cell responses. The authors show that restricted TRAJ33-bearing T cells have Tfh cell phenotype, but it is unlikely that restricted TRAJ33-bearing T cells provide a cognate help to B cells autoreactive to pancreatic protein antigens. Would it be possible that abnormal Tfh cells are generated and responsible for the disease? Would CD3e^{-/-} mice develop pancreatitis if they received T cell population without restricted TRAJ33-bearing T cells?

Minor points:

#1 Many figure legends are missing important information, such as timing when tissues are harvested for analysis in Fig. 1C 1J 1K 1L 1M 2A-C, 2E 1F 2I 2J 2K 2M 3A-K 4B-L 6E-J S1 and the percentages are shown among what cell population in Fig 4G etc...Are Data shown as mean +/-sd?

#2 The orders of panels in Figures should be placed sequentially. They should also appear sequentially in the results.

#3 It is not clear to me if the pancreatic islet is intact in Fig 1C. Perhaps the intact pancreatic islet can be indicated by arrows in the images.

#4 Fig 1L. why is the CD5 expression changed over time in Bcl11b Δ iThy mice?

#5 Were plasma cells infiltrating into the pancreas of Bcl11b Δ iThy mice? Was this an MR1 dependent?

#6 Fig 4G. What about GC B cell and Tfh cell populations in Bcl11b Δ iThy MR1^{-/-} mice?

#7 Fig 5C. The frequencies of non-TRAV1-2 / TRAJ33+ cells among CD69+ cells seem comparable between healthy and AIP subjects. The percentages of total and CD69 positive TRAV1-2/non-TRAV1-2 cells should be shown.

#8 Fig 5E. shows the percentages of CD69^{high} T cells among Ac-6-FP tetramer + cells. Why are most of the values higher than 100?

#9 Fig 5F. Do Ac-6-FP tetramer + cells produce IL-21?

Reviewer #1 (Remarks to the Author):

Summary

In addition to MR1-5-OP-RU reactive MAIT cells, an increasing number of reports are emerging of other MR1-reactive T cells. However, their protective or pathogenic immune roles are unknown. Here, the authors describe the development of pancreatitis in mice where *Bcl11b* is ablated in immature thymocytes. They show the accumulation of pathogenic CD4+ MR1-reactive T cells in these mice, which appear to drive the disease progression. Based on reactivity to fractionated tissue homogenates, followed by mass spec analysis the authors identify riboflavin as an antigen presented by MR1 and recognized by the aberrant T cells.

This paper describes potentially exciting findings and contains several aspects that will be of interest to the field. However, there are many unanswered questions that would need to be addressed prior to publication. In particular, the later parts of the story around antigen identification would need significant further work to confirm what could be very interesting and important findings. Specific comments are provided below.

We sincerely thank the reviewer for constructive comments.

Major comments

1) The first few figures show an interesting phenotype in the *Bcl11b* deficient mice. One issue is the small numbers of mice presented for several of the figures (e.g. parts of figures 1, 2, 6 show only n=3-5 mice). The authors should ensure that these are representative of several repeats using age-matched mice. Fig 2B (showing no CD1d-aGC or MR1-5-OP-RU reactive cells in the spleens shows only one mouse for each).

We agree with the reviewer's suggestion that experimental repeats are important for confirming the phenotype of *Bcl11b*^{ΔiThy} mice. In accordance with the reviewer's suggestion, we have clearly described 2 to 3 repeats of these experiments using age-matched mice (Figure legends of Fig.1, Fig. 2, and Fig. 6).

2) The role of MR1 in thymic selection vs peripheral reactivity is topical in the MAIT/MR1T cell field. It is possible that MR1T cells (other than MAIT cells) may be selected on MHC molecules and cross-reactive on MR1. However, this study shows that the aberrant T cells

did not develop when crossed to MR1^{-/-} mice. I feel the authors could build on this part of the story, which would be of great interest to the field.

We thank the reviewer for this interesting suggestion. To determine the requirement of MHC molecules for the development of T cells in *Bcl11b*^{ΔiThy} mice, we further introduced MHCII-deficient alleles (*Bcl11b*^{ΔiThy} × *H2-Ab1*^{-/-}) and found that splenic CD4⁺ T cells were decreased but still detected (shown below). This finding supports the idea that MHC may contribute to the selection process or peripheral maintenance of these T cells. However, as this hypothesis needs to be addressed by further experiments, we did not include these data in the current manuscript.

3) The aberrant T cells don't accumulate when mice are crossed with MR1^{-/-}. The authors state they are not MAIT cells based on lack of PLZF expression. A recent publication (Drashansky et al. *iScience* 24(4): 102307, April 2021) showed that MAIT cell development is impaired in *Bcl11b* deficient mice, particularly MAIT-17 cells due to the blockade of PLZF and RORγt, needed for their development. This study should be considered and referenced. In accordance with the reviewer's recommendation, we have added the relevant description (page 9, lines 1–5) and the reference (Drashansky et al., 2021) (Ref. 13) in the revised manuscript.

4) The authors use the lack of TRAV1, lack of PLZF and lack of reactivity of one clone to MR1-5-OP-RU as evidence that these cells are not MAIT cells. Although not essential, it would be interesting to understand the location of these cells within various organs – do they have a mucosal distribution? What is their chemokine receptor expression?

In accordance with the reviewer's recommendation, we have analyzed the tissue distribution of these cells. Since definitive markers were not available, we enriched T cells and performed bulk TCR sequencing to ascertain the frequency of *Trav1*⁻*Traj33*⁺ cells that comprise MR1T cells as shown below. From this analysis, we have not been

able to observe the clear mucosal distribution of these cells. We believe that the identification of a definitive marker will clarify this issue.

Following the reviewer's question, we have investigated chemokine receptor expressions in *Traja33*⁺ T cells of *Bcl11b*^{ΔiThy} mice. The expressions including CCR6 (homing to the intestine), CCR7 (homing to the lymph node), CCR9 (homing to the intestine) and CCR10 (homing to the skin), were not different between *Traja33*⁺ T cells and other T cells (shown below). Thus, we have not been able to find a particular gene expression signature related to the mucosal distribution.

5) There are many questions around the later part of the study looking at the identification of the antigen.

i) The authors show a structure of a TCR superimposed on MR1-6-FP, but seem to state later that this TCR is reactive towards MR1 presenting riboflavin. Does MR1 refold with riboflavin? Alternatively can MR1-riboflavin modelled with the TCR in a similar way to the

MR1-6-FP structure? If riboflavin sits differently in the cleft the TCR docking with MR1-6-FP is not informative here.

In accordance with the reviewer's question, we have tested whether MR1 refolds with riboflavin similar to the case of 6-FP (Corbett et al., 2014). We have successfully detected the peak corresponding to the MR1-6FP complex, which is consistent with the previous studies; however, MR1 did not refold with riboflavin, probably because of the weak affinity of riboflavin to MR1, as previously reported (Harriff. et al., 2019). Nevertheless, riboflavin agonistic activity to clonotype #1 was blocked by Ac-6-FP, as shown in the original Fig. 7 c, suggesting that riboflavin binds to MR1 in a similar manner to the MR1-6-FP structure.

ii) Although clonotype 1 was most frequent, many other clonotypes were seen (Fig 3A). Were these also reduced on an MR1^{-/-} background? Were these reactive with MR1-riboflavin? Key results supporting riboflavin as an antigen in Fig 7F, G need further validation. Is the upregulation of surface MR1 specific (an MHC I control could be used). What do the error bars represent in these figures?

As indicated by the reviewer, we are also interested in whether other clonotypes are decreased on an *Mr1*^{-/-} background. We have demonstrated that several expanded clonotypes in *Bcl11b*^{ΔiThy} mice were also decreased on an *Mr1*^{-/-} background (shown below).

However, we have not detected evident reactivity of these TCRs with riboflavin, presumably since these were *Traj33*-negative.

According to the reviewer's suggestion, we have analyzed MHC class I expression as a control. Riboflavin did not induce MHC class I upregulation (shown below).

Furthermore, following the reviewer's suggestion, we have added the sentence "Data are expressed as mean \pm SEM." in the Figure legends section of the revised manuscript (page 66, lines 13–14).

iii) From an evolutionary perspective, detection of riboflavin precursor compounds (ie 5-OP-RU) makes sense, whereas the detection of riboflavin may lead to autoimmunity. The authors should speculate on the physiological relevance of this in normal mice, particularly given the response (Fig 7G) is much less sensitive than that of MAIT TCR for 5-OP-RU.

We thank the reviewer for suggesting this interesting Discussion. Under normal condition, riboflavin does not cause autoimmunity, presumably because of its low affinity and low frequency of reactive clonotypes. Rather, this machinery might speculatively sense disequilibrium of metabolites that reflect tissue damage, followed by transregulation of other T cell subsets. Therefore, we have added our speculation in the Discussion section of the revised manuscript (page 20, lines 7–9).

6) Has the reduction in MAIT cells in AIP patients been reported previously? The authors note that MAIT cells are reduced and MR1T cells are present at normal levels in these patients. They then remark that the ratio of these cells is altered (Fig. 5C), which seems obvious. Numbers rather than ratio should be shown for MR1-Ac-6-FP reactive cells from multiple donors.

We thank the reviewer for this comment. To the best of our knowledge, there have been no reports demonstrating the decrease of MAIT cells in AIP patients.

We agree with the reviewer's suggestion that absolute numbers are the best way to demonstrate the populational changes of MAIT and MR-reactive/restricted T cells. However, we have not been able to compare the absolute numbers of these populations, as total cell numbers in PBMC were different in each individual. Instead, we have added the frequencies of MAIT (*TRAV1-2/TRAJ33*⁺) and non-MAIT (non-*TRAV1-2/TRAJ33*⁺) cells in revised Fig. 5 c, as shown below.

7) In both mice and humans the authors refer to the balance of MR1T cells compared to MAIT cells or other T cells being important for pathology. However, to my knowledge no regulatory function has been attributed to MAIT cells. Do the authors consider the presence of MAIT cells decreases the ability of the pathogenic cells to expand (niche effect)? This could perhaps be addressed in the mice by the transfer of titrated numbers of cells.

We thank the reviewer for pointing out this important issue. Indeed, no clear regulatory function has been attributed to MAIT cells, although it is proposed in some reports (Koay et al., 2016; Koay et al., 2019). Instead, we were also interested in the possibility, as the reviewer suggested, that multiple MR1-restricted/reactive T cells are mutually competing both in the thymus and periphery on the MR1 niche. We therefore agree the reviewer's suggestion and transferred titrated numbers of MAIT cells (1×10^5 and 2×10^4) to *Bcl11b*^{ΔiThy} mice; however, this treatment did not improve the disease symptoms. For that

reason, we have weakened this possibility in the Abstract section (page 3, lines 13–14) and the Results section of the revised manuscript (page 10, lines 2–7).

8) Was any attempt made to identify the other m/z compounds? It seems unlikely that the one known compound would be identified as the antigen. It is possible that other related compounds are more potent antigens or MR1 ligands. In this case the techniques used to identify 5-OP-RU (refolding MR1 in the presence of the positive fraction) could be used for ligand identification.

As the reviewer mentioned, we identified the other unannotated m/z compounds as candidates of antigens. We have tried to purify these compounds on a large scale to conduct NMR spectrometry. Additionally, we have attempted MR1 refolding in the presence of the positive fraction. However, we have not obtained correctly folded MR1 neither by adding this fraction nor by adding riboflavin (shown below). Thus, reporter cells might be more sensitive than refolding MR1 for the identification of low affinity antigens.

9) It is unclear from the data shown whether the clonotype 1 TCR recognizes MR1-riboflavin or MR1 independent of antigen (as has previously been described for other MR1-reactive T cells). In this case riboflavin may not be an antigen but may cause reactivity by non-specifically upregulating MR1. Does the clone also respond to MR1 in the presence of Ac-6-FP? The ternary structure as well as further functional analysis would be important here in understanding the specificity. In the case that this clone is not recognizing a specific antigen, how does this affect the interpretation of the TCR extending into the Ag-binding cleft as modelled for MR1-6-FP?

We apologize for our unclear explanation in Fig. 7 c, g. The activation of reporter T cells expressing clonotype #1 TCR was riboflavin-dependent and was inhibited in the presence of Ac-6-FP (original Fig. 7 c) and anti-MR1 mAb (original Fig. 7 g).

Minor comments

1) The naming of MR1-reactive T cell subsets is still unclear in the field. The authors should consider their use of “MR1-restricted” vs “MR1-reactive” since it is possible that some MR1-reactive T cells are cross-reactive.

We agree that the naming is important. “MR1-restricted” and “MR1-reactive” were distinguished in our manuscript depending on the context. As for the expanded T cells in *Bcl11b*^{ΔiThy} mice, we have carefully avoided using “MR1-restricted”, because we did not prove that these T cells are developed in an MR1-dependent manner in the thymus. However, we observed some evidence that these T cells “reacted” with MR1, we used “MR1-reactive” in this manuscript.

2) Change of tense in abstract and in first section of results.

According to the reviewer’s comment, we have corrected tenses in the Abstract section (page 3, lines 5–7) and the first section of Results (page 6, line 3) in the revised manuscript.

3) Fig.1.

i) B says n=5, but it seems only 4 mice are shown.

We apologize for the incorrect information. We have corrected it in the revised manuscript (page 57, line 4).

ii) Naming/positioning of panels D-F should be reconsidered (also for other figures).

According to the reviewer’s comment, we have changed the positioning of Fig. 1 d-f, Fig. 2, Fig. 3 and Fig. 5 in the revised manuscript.

iii) What age are the mice shown in C-E?

We have added the information in the Figure legends section of the revised manuscript (pages 57–58).

iv) In F, further explanation of the glucose levels should be added to the text or legend. The authors state that mice did not develop diabetes, but do show a statistical difference (albeit in only 3-5 mice)

We apologize for our misleading expression. The blood glucose levels in *Bcl11b*^{ΔiThy} mice were decreased rather than increased, as found in diabetes. This suggests that β cells might not be damaged in these mice. These tendencies were also observed in human AIP patients (Esposito et al., 2020). We have rewritten this sentence to clarify this point in the revised manuscript (page 6, lines 10–12).

4) Sup fig 1B. histology comparison of WT and Bcl11biThy mice should be shown on the same scale.

Following the reviewer's comment, we have changed the images with the same scale in the revised supplementary Fig. 1 **b**.

5) 5-OP-RU is not listed in the methods, but is presumably generated from 5-A-RU and methylglyoxal, which are listed. The method for this and the validation of concentration should be shown. Previous studies have shown this reaction to be inefficient, so if the stated concentrations for 5-OP-RU on Figure 7H are really indicating 5-A-RU used in the reaction this should be stated, since this is likely an overestimate of 5-OP-RU concentration.

We agree with the reviewer's comment that this reaction is inefficient. Currently, the definitive amount of 5-OP-RU in our experimental setting is difficult to state; thus we have carefully described our procedure of 5-OP-RU conversion in the Methods section (page 28, lines 11–12) and clearly shown that 5-OP-RU concentrations in Fig. 7 **h** are indicated under assumption that all 5-A-RU is converted to 5-OP-RU in the Methods section (page 28, lines 12–13) and Figure Legends section of the revised manuscript (page 66, lines 11–12).

6) The methods state that the human MR1-K43A tetramer was used. Why was this chosen over the WT tetramer (or is this an error - since the NIH facility is listed as a source for mouse tetramers)? No source is noted for the K43A tetramer. Was this produced in-house? How does this impact the interpretation since the binding will differ between MR1-WT and MR1-K43A?

We have generated human MR1-K43A tetramer in-house in collaboration with Medical & Biological Laboratories Co., LTD (MBL), and it is now commercially available (Cat. # TS-HMRV2-2). We have added the information in the revised manuscript (page 32, lines 5–6). There is no significant difference in the frequency and MFI of tetramer-positive populations between NIH and MBL tetramers as shown below.

7) Figure 7A. The text (page 17) states that fecal extracts were fractionated and tested for their stimulatory capacity. However, the figure legend refers to fractionated tissue homogenates. Which is correct?

We thank the reviewer for pointing out the mistake. We have corrected the description in the revised manuscript, as fecal extracts were used in this study (page 66, line 1).

Reviewer #2 (Remarks to the Author):

Shibata et al in their manuscript "Symbiotic dependent expansion of MR1-reactive T cells causes pancreatitis in the absence of BCL11b in lymphocytes." Utilizes many techniques to establish the cause of pancreatitis in thymic specific BCL11b knock out mice. The authors beautifully show that there is an expansion of pathogenic non-MAIT MR-1+ ab T cells that induce B cells and pancreatitis in a microbiota dependent manor. Moreover, the authors also evaluate patients with autoimmune pancreatitis to find some similarities with their model. These are potentially exciting findings, but some of the connections and rigor of the experiments are lacking.

We sincerely appreciate the reviewer for raising the important criticisms to improve our study.

Major points:

1. One of the key findings of the manuscript is identification of a novel Mr1+ non MAIT abT cell population. However, very little is done to evaluate this population beyond the prevalence of clone 1 in the population. The authors used advanced techniques such as scRNAseq and CyTOF and yet the analysis of these techniques is very rudimentary, lacking in deep analysis of the data. 33 markers are used in the CyTOF panel, yet there is no comment on what the T cell or B cell population are in the BCL11b k/o mice with the main figure only commenting on the GC B cell and Tfh cell proportions in the ko mice. What are the markers of the T cells and B cell in these k/o mice? Are there unique markers that identify the novel Mr1 population?

We apologize for our unclear explanation for these issues. In CyTOF analysis, T_{FH} cells and GC B cells were annotated based on the expression of PD1/Bcl-6 and GATA3/Bcl6, respectively (original supplementary Fig. 4 c and d). Although the expressions of CD25 and Foxp3 were higher in *Bcl11b*^{ΔiThy} T cells, these trends might have been present due to the activation status of these T cells, as definitive Treg population identified by Foxp3 expression was reduced (shown below).

The proportions of other conventional T cells were lower in *Bcl11b*^{ΔiThy} T cells, as shown in Fig. 2 a. Considering the unique markers for the novel MR1-reactive T cell population, we have reanalyzed the original data and discovered that CD160, CD7, CD74, CD6, CD46 and CD5 were relatively highly expressed in this population (shown below). However, as the contrast was not strong enough to distinguish this population from other T cells, we decided not to use them as unique markers.

What are the overall proportion of the other immune cells in these mice? Similarly, the scRNAseq data is not developed.

We thank the reviewer for raising this important question. scRNAseq analysis was conducted using sorted T cells. Instead, we have carefully analyzed other immune cells using flow cytometric analysis and found that CD11b^{high} Ly6C^{high} myeloid cell frequencies increased with disease progression in *Bcl11b*^{ΔiThy} mice in an MR1-dependent manner (shown below). We are extensively analyzing the pathogenic potential of these cells.

What are the various clusters found, what is driving these clusters.

After clonotype #1 cluster was determined by the loupe V(D)J browser, transcriptome analysis of the cluster was further conducted by the loupe browser. We have described it in the Methods section of the revised manuscript (page 34, lines 13–15).

The tsne plots for the single k/o and Mr1 ko are generated separately. It would be helpful to combine into one analysis.

Following the reviewer's comment, we have combined these data and performed analysis. As shown below, similar to Fig. 3 c and d, we have observed expansion of clonotype #1 and *Traj33*⁺ cells.

With common cluster identification, etc. Where wt mice analyzed? Was clone1 present in wt? What is the shift in the T cell populations between the wt and single ko mice, how is this altered by additional Mr1 ko?

As most of T cells express IL-18R in *Bcl11b*^{ΔiThy} mice, we enriched rarely detected IL-18R⁺ T cells from WT mice and compared gene expressions and TCR repertoire within the similar population. Nevertheless, as WT IL-18R⁺ T cells exhibited distinct phenotypes as assessed by tSNE clustering from *Bcl11b*^{ΔiThy} mice and *Bcl11b*^{ΔiThy}. *Mr1*^{-/-} mice (shown below), we did not use it as a control. Clonotype #1 was not detected within this limited number of WT T cells in this single cell-based analysis (shown below). However, when we further addressed this issue by deep bulk sequencing of TCR α and TCR β chains from all TCR β ⁺ T cells, CDR3-matched TCR α chain of clonotype #1, and similar CDR3-non-matched TCR β of clonotype #1 were detected.

We have described this issue in the Discussion section of our revised manuscript (page 20, lines 15–16).

The scRNAseq data suggests that there is increase in Tfh cells in ko mice, are these markers present in the CyTOF data? or validated in any other way? Flow? Etc.

We apologize for our unclear description. By CyTOF data, typical T_{FH} markers, such as PD-1 and Bcl6, were detected on a subset of T cells in *Bcl11b*^{ΔiThy} mice, as we have shown in the original supplementary Fig. 4 c. Flow cytometric analysis have shown that these T cells express T_{FH}-related molecules, such as CD40L or IL-4 (original Fig. 4 h-j).

2. All antibody subclasses were drastically increased in the periphery not just IgG. Are these clonally expanded? Are they contributing to the disease? B cell combined ko has significantly increased survival arguing that it is the B cells not the T cells themselves that are driving the pathology. Can B cells from these mice be transferred to cause disease?

We transferred IgG from *Bcl11b*^{ΔiThy} mice to the WT recipient, but we did not observe any disease symptoms. We sincerely apologize for the misleading description regarding the role of B cells in the pathology. Survivals of B cell-deficient and IL-4-deficient *Bcl11b*^{ΔiThy} mice were not significantly prolonged compared to *Bcl11b*^{ΔiThy} mice (original supplementary Fig. 4 f). We have corrected our misleading description in the revised manuscript (page 14, lines 6–8).

3. Do the authors believe that the mice are dying from pancreatitis? Why did the authors chose to examine only the pancreas and the large intestine.

We have analyzed various tissues, and among them, the pancreas and large intestine were the two major organs showing severe inflammation. We have also observed inflammation in the lung and lacrimal gland albeit at a lower frequency. We have added these data as a revised supplementary Fig. 1 d (shown below) and its description in the Results section of the revised manuscript (page 6, lines 13–15).

As currently we do not have a definitive answer for the cause of death, we have corrected our misleading description in the Abstract section (page 3, lines 5–7) and in the Introduction section (page 5, lines 6–9) and Results section (page 6, line 2 / page 9, lines 4–5 / page 9, line 16).

4. In the BCL11b k/o + wt transfer experiments, the authors conclude that the restoration of the normal cells precludes expansion of k/o cells. It's possible that the wt cells have a competitive advantage, right? And just outcompete the k/o cells. There are almost no T cells from the k/o mice in the mixed chimera model. Unlike the B cells that are still present. Could it be that these are more prone to apoptosis?

We agree with the reviewer's point. We assume that the correction of abnormal expansion of MR1-reactive T cells by transfer of conventional T cells might be partly caused by their advantage in cell competition. Indeed, *Bcl11b*^{ΔiThy} T cells are prone to apoptosis, which is similar to the previous report of *Bcl11b*^{ACD4} T cells (Albu et al., 2007), as the

reviewer predicted, as shown below. Thus, we have carefully described this in the Results section (page 9, lines 14–16).

5. Fig11 the percent of ab T cells in peripheral blood of ko mice is almost 40% of the PBMCs. Is this due to an overall increase in these cells or proportional decrease in other immune cell populations. What is happening to the other immune populations?

Following the reviewer’s question, we have analyzed time-related changes of B cell frequencies and discovered that they declined in *Bcl11b*^{ΔiThy} mice compared with those of WT mice which were relatively stable over time as shown below.

6. Another key finding of the manuscript is identification of riboflavin as the metabolite driving these Mr1+ T cells, can the authors show that riboflavin is sufficient to drive pathology in the GF mouse experiments?

Actually, we were interested in this possibility. However, we could not test this approach during these review periods due to limitation of the facility. We are now purifying to obtain sufficient amount of this fraction to administer into GF mice in the future study, as other unidentified compounds included in this fraction #54 (Fig. 7 d) might synergistically contribute to the pathogenesis. To include the possibility that MR1-reactive T cells recognize other as-yet unidentified feces components, we have also changed the subtitle of the Results section of the revised manuscript (page 17, line 16).

Minor points:

1. Methods need to be much more developed so that work can be reproduced. For example, Fig 1E is inflammatory score- how is this developed, what is counted.

Similarly, all through out, there is lack of attention to details (on what day were the blood glucose collected, or the lipase? When did the steroid treatment start? Etc).

According to the reviewer's comment, we have added the description in the Methods section (pages 26–27 for steroid treatment / page 38 lines 1–7 for inflammatory score) and the Figure legends section of the revised manuscript (pages 57–66).

2. There is a lack of rigor to the histological and IF data. Most histological images are not quantified. And only one, presumable best example, is shown. For example, Fig1D increase in Sirius red stain between wt and ko, please quantify. Fig2 E and Fig2 K, quantify immunologic score. Figure S1B, quantify thickness. S1C show the colons, etc.

According to the reviewer's suggestion, we have added quantified data for Fig. 1 c, Fig. 1 d, Fig. 2 e and Fig. 2 k in the revised manuscript.

3. Stylistically, please reformat the figures and the text so that both appear in chronological order (so that the sub-figures appear in the figures and within the text in chronological order, a before b, etc). It's challenging to follow in the current state and distracts form the manuscript.

According to the reviewer's suggestion, we have changed the order of Fig. 1, Fig. 2, Fig. 3 and Fig. 5 in the revised manuscript.

4. Nature communication is intended for a general audience, please label portions of the pancreas that the text is referring to.

According to the reviewer's advice, we had added the label in all revised figures.

5. T cell specific abstract misses the other important findings of the manuscript namely B cell and microbiota contribution to the disease

As the reviewer suggested, we wondered whether the contribution of B cells secreting disease-associated autoantibodies was included in this manuscript. However, as we did not observe that B cell depletion did not prolong survival of *Bcl11b*^{ΔiThy} mice (Supplementary Fig. 4 f), we have mentioned pathogenic roles of T cells and contribution

of microbiota (symbiotic bacteria) to the pathogenesis in the abstract (page 3, lines 13–14).

6. Page 6 statement that T cells are required but not B cells is misleading as B cells clearly contribute to the phenotype from the data presented in the manuscript.

We apologize for the misleading description of whether B cells are pathogenic in *Bcl11b*^{ΔiThy} mice. Since we showed that genetic depletion of αβ T cells but not B cells significantly prolonged survival in *Bcl11b*^{ΔiThy} mice, we have corrected this interpretation in the Results section of the revised manuscript (page 14, lines 6–8).

7. It looks like TCRD double ko mice have a worsened phenotype than the BCL11b k/o, are they perhaps ameliorating the phenotype?

We were also interested in this point and speculated that loss of γδ T cells might increase αβ T cell niche that includes pathogenic cells in *Bcl11b*^{ΔiThy} mice. However, as survival between *Bcl11b*^{ΔiThy} mice and *Bcl11b*^{ΔiThy}.*Tcrd*^{-/-} was not statistically significant (original Fig. 1 g), we did not include this issue in the current manuscript.

8. IgG1 deposit pictures in Fig 4D are of poor quality and need to be quantified.

According to the reviewer's suggestion, we have reanalyzed IgG1 deposition and added quantified data in revised Fig. 4 d, as shown below.

9. "The inflamed pancreases of *Bcl11b*^{ΔiThy} mice were infiltrated with IgG1+ B cells and CD4+ αβ T cells." However, the staining is for B220, how do we know these are IgG+?

We thank the reviewer for the question. In revised supplementary Fig. 4 h, we have shown that some B cells in the pancreas of *Bcl11b*^{ΔiThy} mice produce IgG1. Thus, as only some

but not all B cells in the pancreas express IgG1, we have carefully corrected the description in the Results section of the revised manuscript (page 14, lines 8–10).

10. Steroid decreased the levels of IgG, what about other antibodies? Was the increase in steroids started at birth? After pathology developed?

Following the reviewer’s comment, we have analyzed IgA and IgM productions in the serum with or without steroid treatment. Similar to IgG1, IgA but not IgM was decreased, as shown below.

We apologize for lacking the description. We started antibiotic treatment from the age of 5 weeks and added the information in the Methods section of the revised manuscript (pages 26–27).

11. In regards to the scRNAseq analysis of human data. The authors state that the naïve clusters are not well-defined in patients, this is very subjective. Was this data automatically clustered? Are there percent differences between the naïve clusters? What are the clusters? What are their markers? Similarly, authors state that green Tf cluster is well apparent in the disease subjects. This is again very subjective, as it looks like it is only apparent in case #10 with no detectable population in #9 and a very small population in #11. Similarly CD161 cluster is apparent in case #9 and 11.

We thank the reviewer for raising an important criticism for scRNA-seq analysis and apologize for lacking the respective information. Clusters were generated by the K-means cell clustering algorithm in an unbiased manner. We have added the description in the Figure legends section of the revised manuscript (page 63, lines 6–8). We have also added supplementary Table 2 showing significantly upregulated genes ($p < 0.05$) in each cluster to clearly demonstrate the cluster definition. We have changed the description of cluster 1 from “naïve T” to “naïve or central memory (CM) T” in the Results section (page 15, lines 7–9) and the Fig. 5 a of the revised manuscript, as discrimination between naïve T

cells and central memory (CM) T cells is difficult. We have also added the featured genes of each cluster in the Results section (page 15, lines 7–12) and their references (Meckiff et al., 2020; Szabo et al., 2019) (Ref. 25, 26) in the revised manuscript.

What clusters express CD69? How is the figure 5C data generated?

We observed *CD69* expression in various clusters. After identification of *TRAVI-2/TRAJ33*⁺ cells or non-*TRAVI-2/TRAJ33*⁺ cells (Fig. 5 **b**) by the loupe V(D)J browser, *CD69*^{high} cells were further counted to show the percentages of the cells expressing a significantly high level of *CD69*. We have added the description in the Methods section of the revised manuscript (page 34, lines 13–15).

Reviewer #3 (Remarks to the Author):

Shibata K et al demonstrate that mice lacking a T cell-specific transcription factor, Bcl11b in immature thymocytes (Bcl11b Δ iThy mice) develop chronic pancreatitis. Although the pancreatitis was ameliorated by MR1 deficiency or under germ-free condition, MR1-restricted mucosal-associated invariant T(MAIT) cells were not increased in Bcl11b Δ iThy mice. Instead, they found that MR1-restricted TRAJ33-bearing T cells were expanded in these mice. They demonstrated that adoptive transfer of bone marrow from Bcl11b Δ iThy mice into CD3e $^{-/-}$ induced pancreatitis, but co-transfer of bone marrow from Bcl11b Δ iThy and WT mice failed to cause pancreatitis, suggesting that MR1-restricted T cells which were increased by the homeostatic expansion of these cells appear to play roles in the development of pancreatitis. They also showed that activated MR1-restricted T cells are present in the peripheral blood of patients with autoimmune pancreatitis. The pancreatitis in Bcl11b Δ iThy mice was accompanied with elevated autoantibodies against lysates of pancreas, expansion of germinal center B cells and T follicular helper cells. They characterize MR1-restricted TRAJ33-bearing T cells from Bcl11b Δ iThy mice by RNA sequencing and the crystal structure of Trbv7-6-Traj33/Trbv13-2-Trbj1-1 TCR. These MR1-restricted T cells recognized riboflavin presented by MR1 and displayed type 2 T and Tfh cell phenotypes. While this study is interesting, there are major concerns about the data shown.

Major Concerns:

Based on experiments with MR1-deficient mice, MR1-restricted T cells clearly contribute to the pathogenesis of Bcl11b Δ iThy mice. However, major questions concerning the function of MR1-restricted TRAJ33-bearing T cells remain open because they studied this T cell population under extremely unusual condition where T cell development is enormously impaired. In addition, the exact mechanism by which MR1-restricted TRAJ33-bearing T cells contribute to the pancreatitis is unknown. Do these MR1-restricted accumulate in the pancreas and cause inflammation, or activate autoreactive B cells? Would CD3e $^{-/-}$ mice still develop pancreatitis when transferred with restricted TRAJ33-bearing T cells?

We thank the reviewer for raising an important issue. MR1-dependent accumulation of CD4 $^{+}$ T cells to the pancreas in *Bcl11b* Δ iThy mice was detected as shown in original supplementary Fig. 4 g. We have also shown the deposition of IgG1 antibodies in the inflamed pancreas (original Fig.4 d). However, genetic depletion of B cells did not

significantly prolong survival in *Bcl11b*^{ΔiThy} mice (original supplementary Fig. 4 f). Furthermore, we newly generated Bcl6-deficient *Bcl11b*^{ΔiThy} mice (*Bcl11b*^{flox/flox} *Bcl6*^{flox/flox} *Rag1*^{Cre/+} mice) and found that they impaired IgG1 production but still developed chronic inflammation (shown below). These results suggest that autoreactive B cells and antibodies were not prerequisite for chronic inflammation, in agreement with the recent study of AIP patients (Perugino and Stone, 2020).

We also carefully investigated effector cell populations which might contribute to the pathogenesis. The frequencies of SSC^{high} CD11b^{high} Ly6C^{high} myeloid cells were correlated with disease progression in *Bcl11b*^{ΔiThy} mice in an MR1-dependent manner (shown below, upper two panels). Similar population was also infiltrated into the inflamed pancreases (shown below, lower panels). This myeloid population generated in the presence of MR1-reactive T cells is one of the candidates that mediates effector function. However, the molecular mechanisms by which MR1-reactive T cells regulate these myeloid cells remain unknown. We are now extensively investigating this issue.

It is currently difficult to transfer *Traj33*-bearing T cells due to the lack of specific markers for purification. Instead, to address this issue, *CD3e*^{-/-} mice were transferred with splenocytes from SPF *Bcl11b*^{ΔiThy} mice (including *Traj33*-bearing MR1-reactive T cells) and GF *Bcl11b*^{ΔiThy} mice (lacking *Traj33*-bearing MR1-reactive T cells) to compare the phenotype of the recipient mice. Transfer of a *Traj33*⁺ cell-including fraction induced disease symptoms, whereas a cell fraction lacking this population did not have any impact (shown below). Although this evidence did not provide a direct answer to the reviewer, the causal effect of this population was verified by this experiment.

If restricted TRAJ33-bearing T cells are activated by recognizing riboflavin generated by symbiotic microbes, why is the main disease in *Bcl11b*^{ΔiThy} mice is pancreatitis, not colitis? We apologize for the misleading description. In fact, we have also observed chronic inflammation in the large intestine. Therefore, we have added the data in revised supplementary Fig. 1 d as shown below. One possibility for this tissue-restricted phenotype would be that the pancreas and intestine are anatomically connected via ducts through which mobile MR1-reactive T cell antigens can be delivered (Adolph et al., 2019) (Ref. 58). This speculative description was included in the Discussion section of the revised manuscript (page 25, lines 4–7).

The generation of autoantibodies against pancreatic lysates was accompanied by enhanced germinal center B cell and Tfh cell responses. The authors show that restricted TRAJ33-bearing T cells have Tfh cell phenotype, but it is unlikely that restricted TRAJ33-bearing T cells provide a cognate help to B cells autoreactive to pancreatic protein antigens. Would it be possible that abnormal Tfh cells are generated and responsible for the disease?

We were also interested in this point and agreed that it is less likely considering immunological principle. However, as some reports previously showed the non-cognate and bystander B cell help by innate-type T cells such as NKT cells and MAIT cells (Bennett et al., 2017; Chang et al., 2011; King et al., 2011), we have carefully discussed this minor possibility that MR1-restricted *Traj33*-bearing T cells might provide a non-cognate help to B cells against pancreatic antigens in a MR1-dependent manner (page 22, lines 6–7).

In accordance with the reviewer’s question, we have examined the contribution of abnormal T_{FH} cells by establishing mice in which both *Bcl11b* and *Bcl6* were deleted (*Bcl11b*^{flox/flox} *Bcl6*^{flox/flox} *Rag1*^{Cre/+} mice). IgG production was dramatically reduced in *Bcl6*-deficient *Bcl11b*^{ΔiThy} mice (left), whereas disease symptoms were still observed (right), suggesting that the abnormal T_{FH} cells is not necessarily required for the pathogenesis of *Bcl11b*^{ΔiThy} mice (shown below).

Minor points:

#1 Many figure legends are missing important information, such as timing when tissues are harvested for analysis in Fig. 1C 1J 1K 1L 1M 2A-C, 2E 1F 2I 2J 2K 2M 3A-K 4B-L 6E-J S1 and the percentages are shown among what cell population in Fig 4G etc...Are Data shown as mean +/-sd?

According to reviewer’s suggestion, we have added the information of the timing for tissue collection in the Figure legends section (pages 57–66) of the revised manuscript. We have also added the sentence “Data are expressed as mean ± SEM” in the Figure

legend section of the revised manuscript (page 63, line 1 / page 65, line 9 / page 66, line 13).

#2 The orders of panels in Figures should be placed sequentially. They should also appear sequentially in the results.

Following the reviewer's comment, we have changed the positioning of Fig. 1, Fig. 2, Fig. 3 and Fig. 5 in the revised manuscript.

#3 It is not clear to me if the pancreatic islet is intact in Fig 1C. Perhaps the intact pancreatic islet can be indicated by arrows in the images.

According to the reviewer's suggestion, we have added the data demonstrating intact islet with an indication of an arrow in revised Fig. 1 c.

#4 Fig 1L. why is the CD5 expression changed over time in *Bcl11b*^{ΔiThy} mice?

We apologize for the mistake. We have corrected x-axis values that showed the comparable CD5 expression level between WT and *Bcl11b*^{ΔiThy} mice.

#5 Were plasma cells infiltrating into the pancreas of *Bcl11b*^{ΔiThy} mice? Was this an MR1 dependent?

According to the reviewer's suggestion, we have examined the presence of CD138⁺ plasmablasts in the pancreas of WT and *Bcl11b*^{ΔiThy} mice. Although the infiltration of CD138⁺ cells observed in *Bcl11b*^{ΔiThy} mice appeared to be decreased in *Bcl11b*^{ΔiThy} × *Mr1*^{-/-} mice in some pancreases (shown below), we could not observe this tendency in all mice. As we did not reach to the definitive conclusion by these experiments, we did not include these data in the current manuscript.

#6 Fig 4G. What about GC B cell and Tfh cell populations in *Bcl11b*^{ΔiThy} MR1^{-/-} mice?

Following the reviewer's comment, we have examined the PNA⁺ GC-like structure in the pancreatic LN of *Bcl11b*^{ΔiThy} × *Mr1*^{-/-} mice. GC-like structure was smaller in *Bcl11b*^{ΔiThy} × *Mr1*^{-/-} mice than *Bcl11b*^{ΔiThy} mice (shown below), suggesting that the development of GC B cells was impaired.

We have reanalyzed scRNA-seq data of all T cells including non-expanded T cells to see T_{FH}-like cell population and detected Cluster 2 expressing *Cxcr5*, *Il21* and *Pdcd1* in both *Bcl11b*^{ΔiThy} and *Bcl11b*^{ΔiThy} × *Mr1*^{-/-} mice (shown below), suggesting that MR1-independent T_{FH}-like cells are also developed in the absence of Bcl11b.

#7 Fig 5C. The frequencies of non-TRAV1-2 / TRAJ33⁺ cells among CD69⁺ cells seem comparable between healthy and AIP subjects. The percentages of total and CD69 positive TRAV1-2/non-TRAV1-2 cells should be shown.

According to the reviewer's suggestion, we have added the data of frequencies of indicated populations in revised Fig. 5 c, as shown below.

#8 Fig 5E. shows the percentages of CD69^{high} T cells among Ac-6-FP tetramer + cells. Why are most of the values higher than 100?

We sincerely apologize for our incorrect presentation. The value in the y-axis is not in percentage, but the mean fluorescence intensity (MFI). Therefore, we have corrected it in the revised manuscript.

#9 Fig 5F. Do Ac-6-FP tetramer + cells produce IL-21?

Following the reviewer's comment, we have reanalyzed the *IL21* expression in non-*TRAVI-2* / *TRAJ33*⁺ cells using scRNA-seq data. However, we have not been able to detect the significantly high expression of the gene in the cell populations of both AIP patients and healthy donors. Therefore, we concluded that IL-21 is not spontaneously produced in these T cells. Consequently, we assume that IL-21 might be detected upon TCR stimulation of these cells; however, we have not been able to conduct the experiment due to the limitation of the availability of patient samples.

References

- Adolph, T.E., Mayr, L., Grabherr, F., Schwarzler, J., and Tilg, H. (2019). Pancreas-Microbiota Cross Talk in Health and Disease. *Annu Rev Nutr* 39, 249-266. 10.1146/annurev-nutr-082018-124306.
- Albu, D.I., Feng, D., Bhattacharya, D., Jenkins, N.A., Copeland, N.G., Liu, P., and Avram, D. (2007). BCL11B is required for positive selection and survival of double-positive thymocytes. *J Exp Med* 204, 3003-3015. 10.1084/jem.20070863.
- Bennett, M.S., Trivedi, S., Iyer, A.S., Hale, J.S., and Leung, D.T. (2017). Human mucosal-associated invariant T (MAIT) cells possess capacity for B cell help. *J Leukoc Biol* 102, 1261-1269. 10.1189/jlb.4A0317-116R.
- Chang, P.P., Barral, P., Fitch, J., Pratama, A., Ma, C.S., Kallies, A., Hogan, J.J., Cerundolo, V., Tangye, S.G., Bittman, R., et al. (2011). Identification of Bcl-6-dependent follicular helper NKT cells that provide cognate help for B cell responses. *Nat Immunol* 13, 35-43. 10.1038/ni.2166.
- Corbett, A.J., Eckle, S.B., Birkinshaw, R.W., Liu, L., Patel, O., Mahony, J., Chen, Z., Reantragoon, R., Meehan, B., Cao, H., et al. (2014). T-cell activation by transitory neoantigens derived from distinct microbial pathways. *Nature* 509, 361-365. 10.1038/nature13160.
- Drashansky, T.T., Helm, E.Y., Curkovic, N., Cooper, J., Cheng, P., Chen, X., Gautam, N., Meng, L., Kwiatkowski, A.J., Collins, W.O., et al. (2021). BCL11B is positioned upstream of PLZF and RORgammat to control thymic development of mucosal-associated invariant T cells and MAIT17 program. *iScience* 24, 102307. 10.1016/j.isci.2021.102307.
- Esposito, I., Hruban, R.H., Verbeke, C., Terris, B., Zamboni, G., Scarpa, A., Morohoshi, T., Suda, K., Luchini, C., Klimstra, D.S., et al. (2020). Guidelines on the histopathology of chronic pancreatitis. Recommendations from the working group for the international consensus guidelines for chronic pancreatitis in collaboration with the International Association of Pancreatology, the American Pancreatic Association, the Japan Pancreas Society, and the European Pancreatic Club. *Pancreatology* 20, 586-593. 10.1016/j.pan.2020.04.009.
- Kaji, T., Ishige, A., Hikida, M., Taka, J., Hijikata, A., Kubo, M., Nagashima, T., Takahashi, Y., Kurosaki, T., Okada, M., et al. (2012). Distinct cellular pathways select germline-

encoded and somatically mutated antibodies into immunological memory. *J Exp Med* *209*, 2079-2097. 10.1084/jem.20120127.

King, I.L., Fortier, A., Tighe, M., Dibble, J., Watts, G.F., Veerapen, N., Haberman, A.M., Besra, G.S., Mohrs, M., Brenner, M.B., and Leadbetter, E.A. (2011). Invariant natural killer T cells direct B cell responses to cognate lipid antigen in an IL-21-dependent manner. *Nat Immunol* *13*, 44-50. 10.1038/ni.2172.

Koay, H.F., Gherardin, N.A., Enders, A., Loh, L., Mackay, L.K., Almeida, C.F., Russ, B.E., Nold-Petry, C.A., Nold, M.F., Bedoui, S., et al. (2016). A three-stage intrathymic development pathway for the mucosal-associated invariant T cell lineage. *Nat Immunol* *17*, 1300-1311. 10.1038/ni.3565.

Koay, H.F., Gherardin, N.A., Xu, C., Seneviratna, R., Zhao, Z., Chen, Z., Fairlie, D.P., McCluskey, J., Pellicci, D.G., Uldrich, A.P., and Godfrey, D.I. (2019). Diverse MR1-restricted T cells in mice and humans. *Nat Commun* *10*, 2243. 10.1038/s41467-019-10198-w.

Meckiff, B.J., Ramirez-Suastegui, C., Fajardo, V., Chee, S.J., Kusnadi, A., Simon, H., Eschweiler, S., Grifoni, A., Pelosi, E., Weiskopf, D., et al. (2020). Imbalance of Regulatory and Cytotoxic SARS-CoV-2-Reactive CD4(+) T Cells in COVID-19. *Cell* *183*, 1340-1353 e1316. 10.1016/j.cell.2020.10.001.

Perugino, C.A., and Stone, J.H. (2020). IgG4-related disease: an update on pathophysiology and implications for clinical care. *Nat Rev Rheumatol* *16*, 702-714. 10.1038/s41584-020-0500-7.

Szabo, P.A., Levitin, H.M., Miron, M., Snyder, M.E., Senda, T., Yuan, J., Cheng, Y.L., Bush, E.C., Dogra, P., Thapa, P., et al. (2019). Single-cell transcriptomics of human T cells reveals tissue and activation signatures in health and disease. *Nat Commun* *10*, 4706. 10.1038/s41467-019-12464-3.

REVIEWER COMMENTS

Reviewer #1 (Remarks to the Author):

The authors have addressed several of the concerns, including presenting a significant amount of new data and many modifications to the manuscript. However, concerns remain regarding both the presentation of data, and the conclusions drawn by the authors.

Major comments (previous numbers indicated)

1) There is still a concern about mouse numbers. For example, Fig 1b shows survival of WT (5 mice) and *bcl1* (4 mice). The authors state these Data are combined from (b) three and (f, g) two independent experiments. So, the total mice are still only 5 and 4 (it is unclear how these are combined from 3 experiments, which would only then have 1-2 mice per group). Surely the authors have many more mice they could easily analyse for survival over time. Fig 1f and g are better with 8-10 mice per group.

In fig 6 a-c. The legend states that there were 3 SPF housed *Bcl11b* mice. This likely explains the large error bars in c, and shows the need for further repeats (this appears to have been done as the legend states Data are representative of 2 experiments). In B there appears to be more than the stated 3 mice in this group (black line shows at least 6 mice).

5)-9)

I remain unconvinced that the conclusions the authors draw are supported by the data (several points are outlined below):

The blocking with Ac-6-FP (Fig 7c) is shown with fraction #54, not with RF as stated in the authors' response.

The MR1 upregulation with riboflavin (Fig 7f) uses an isotype control only in the absence of RF. The antibody and fluorochrome used to detect MR1 are not described. As riboflavin itself is fluorescent the signal here could be artefactual without showing the correct controls.

MR1 blocking of T cell reporter activation by RF (Fig 7g) could still indicate ligand independent MR1 recognition. Ac-6-FP blocking should be done in this experiment.

The data presented on the other clonotypes, which are also reduced on an MR1^{-/-} background would be important to include in the manuscript.

The conclusions around RF being the antigen should be modified given the lack of concrete evidence for this.

E.g. Abstract, line 11-13 "...identified the antigen of the TCR expressed by pathogenic T cell..." should be modified. RF is a candidate antigen that may be recognized by one clone (albeit the most frequent). It is also not clear that this clone is responsible for the pathology as several other clonotypes were also observed.

Discussion (page 20, line 5): "Given that RF activated T cells in *Bcl11b* Δ *Thy* mice, this metabolite must regulate MR1-reactive T cells as both an agonist and antagonist depending on the TCR repertoire". There is no evidence for RF activating the cells in the mice (only the presence of the microbiome). There also appears to be no evidence for the latter part of the sentence.

Minor comments (previous)

4) Sup fig 1B. There is now a major issue with this figure.

To clarify, by "scale" I meant that the same magnification should be used for WT and *Bcl11b* samples. The WT sample in the new Supp figure appears to be the same image from the previous version of the figure, which is now stretched in the horizontal dimension. It is thus unclear if the two samples are of comparable magnification. Please correct.

Further comments on changes made to the revised manuscript.

(major)

1) According to data shown Fig 1a, the mice did not lose weight as described in the results (Page 6, Line 4). From this data, it would be more correct to say that they did not gain weight like their WT counterparts. However, this data is inconsistent with that shown in Fig 6a where the SPF housed mice gain weight normally, but then lose weight sharply from around 10 weeks of age. Can the authors explain this inconsistency?

2) supp Fig 1d. It is impossible to interpret this figure. What do the symbols represent?

3) Supp Fig 2e. Why are the cells from Bcl11b-thy mice only shown for CD4+ cells? There is no mention of CD4 in the manuscript prior to this point. Also the Figure legend states that WT cells are shown, but they are not in the figure.

4) Figure 2h. Page 9, line 15: text describes expansion of Bcl11b Δ iThy cells, figure shows CD4 T/B cells (%), but the use of CD4 here is not explained. Also CD4 T/B cells (%) does not make sense. Page 9, line 15 "...partly due to the survival disadvantages." This is not shown. If it is speculation that this may be the cause this should be stated as a possible cause not definitively as written here.

5) Page 9, Line 16: chronic inflammation induced by Bcl11b Δ iThy BM cells was eliminated by co-transfer with WT BM cells (Fig 2k, Supp Fig 2f). The supp figure shows only one representative image for each. Fig 2k appears to show 2 mice, but there is no statistical test shown. Was this significant?.

6) Apologies for not picking this up earlier, but there is an extra residue in the CDRa (Fig 3f) compared to typical MAIT TCRs. Critically, this would shift the Tyr95 residue (thus no longer at position 95), impacting interpretation of the crystal structure virtual superposition. Indeed, Supp Fig 3e clearly shows a different position for the Tyr, while the authors' description that it points towards the cleft remains true. Thus, the text should be modified to accommodate this finding, which is actually very interesting.

(minor)

Abstract. Line 8-9 "a skewed receptor" or "skewed receptors"

Line 10. "...type 2 MAIT cells have not been reported". There have been reports of type 2 cytokine production by MAIT cells, so this statement should be adjusted.

Abstract. Final sentence is not needed as the result was stated above regarding germ-free conditions. Page 6, line 8. "As" is not needed in this sentence.

Page 10, line 2: Fig 2l and m. Only 3-4 mice per group is insufficient. There is no mention of repeat experiments for l. In m, a paired test could be used for the donor/recipient cells within the same mouse.

Reviewer #2 (Remarks to the Author):

All of my comments were sufficiently addressed.

Reviewer #3 (Remarks to the Author):

This paper by Shibata et al. provided interesting study on non-MAIT MR1-restricted T cells, but there are issues that must be addressed to draw conclusions regarding the function and contribution of these cells in autoinflammatory disorders.

1. The authors claim that MR1-restricted T cells including clonotype #1 T cells are responsible for

pancreatitis because they are accumulated in the pancreas. However, they only showed that “CD4+ cells” observed in the pancreas of Bcl11b Δ iThy mice were disappeared by MR1 deficiency in these mice (Figure 4d). The pathology in Bcl11b Δ iThy mice seems dependent on MR1 and T cells, but data presented here do not demonstrate that MR1-restricted clonotype #1 T cells are actually responsible for the disease in Bcl11b Δ iThy mice unless the dependency is confirmed by experiments with as the adoptive transfer of these cells or agonistic agents for these cells, or by using mice deficient with this specific T cell population.

The current data in this paper suggest that any T cells restricted by MR1 may be responsible for the disease in Bcl11b Δ iThy mice. It is well recognized that MR1 deficiency influences microbial composition (Varelias A JCI 2019; Smith AD PLOS ONE 2019), and disease phenotype in MR1 $+/+$ mice co-housed with MR1 $-/-$ mice is often different from that in wild-type mice obtained from a vendor. Thus, to confirm the contribution of MR1-restricted T cells, Bcl11b Δ iThy x Mr1 $-/-$ mice should be compared with co-housed littermate Bcl11b Δ iThy mice (Fig 2d, 2e, 2f, 3a-j, 4d-f). To demonstrate the contribution of MR1-restricted T cells in the pathogenesis, Bcl11b Δ iThy x Mr1 $-/-$ mice (as well as co-housed littermate Bcl11b Δ iThy mice) should be also analyzed for experiments in Fig. 4c, 4g-j, 4k, 4l, 6 a-k).

2. The authors demonstrated inflammation in the pancreas and colitis and hypergammaglobulinemia (including IgM, IgG, IgA, and IgE) in Bcl11b Δ iThy mice. However, in human IgG4-related disease, the colitis is very rare (0.8%) and the exclusion criteria includes comorbidities including inflammatory bowel diseases (2019 ACR/EULAR Classification criteria for IgG4-related disease). In addition, there is no elevation of IgM and IgA in IgG4-related disease (Rheumatology 2015. 54: 1982–1990). Thus, the disease in observed in Bcl11b Δ iThy mice should be considered different from IgG4-related disease and Figure 5 should be excluded from the paper. I think Figure 5 is not informative because age and sex of healthy and AIP specimens appear to be not matched. In addition, non-MAIT MR1-restricted T cells were identified based on the recognition of Ac-6-Fc which is not recognized by clonotype #1 T cells.

Reviewer #1 (Remarks to the Author):

The authors have addressed several of the concerns, including presenting a significant amount of new data and many modifications to the manuscript. However, concerns remain regarding both the presentation of data, and the conclusions drawn by the authors.

Major comments (previous numbers indicated)

1) There is still a concern about mouse numbers. For example, Fig 1b shows survival of WT (5 mice) and *bcl1* (4 mice). The authors state these Data are combined from (b) three and (f, g) two independent experiments. So, the total mice are still only 5 and 4 (it is unclear how these are combined from 3 experiments, which would only then have 1-2 mice per group). Surely the authors have many more mice they could easily analyse for survival over time. Fig 1f and g are better with 8-10 mice per group.

We apologize for our incorrect description in the legend of Fig. 1 b, f and g. In Fig. 1 b, we show representative results from three independent experiments as shown below.

In fig 6 a-c. The legend states that there were 3 SPF housed *Bcl11b* mice. This likely explains the large error bars in c, and shows the need for further repeats (this appears to have been done as the legend states Data are representative of 2 experiments). In B there appears to be more than the stated 3 mice in this group (black line shows at least 6 mice).

We apologize for our mistaken statement showing the incorrect numbers of SPF *Bcl11b*^{ΔiThy} mice in the original Fig. 6 a-c (revised Fig. 5 a-c). As pointed out by the reviewer, we agree that the number of mice in the original Fig. 6 c was wrong. Accordingly, we have carefully checked them in the original Fig. 6 a-c and not show the corrected data in the revised Fig. 5 a-c in the revised manuscript (page 59, line 6).

5)-9)

I remain unconvinced that the conclusions the authors draw are supported by the data

(several points are outlined below):

The blocking with Ac-6-FP (Fig 7c) is shown with fraction #54, not with RF as stated in the authors' response.

We apologize for our incorrect description in the response letter. We have now added new data in the revised Fig. 6 g as shown below.

The MR1 upregulation with riboflavin (Fig 7f) uses an isotype control only in the absence of RF. The antibody and fluorochrome used to detect MR1 are not described. As riboflavin itself is fluorescent the signal here could be artefactual without showing the correct controls.

In accordance with the reviewer's comment, we performed an additional experiment using an isotype control mAb to appropriately evaluate surface MR1 expression (shown below). Accordingly, we have added the data in revised Fig. 6 f. In addition, we clearly described the antibody and fluorescence label used in this experiment in the Figure legends section of our revised manuscript (page 61, line 8).

MR1 blocking of T cell reporter activation by RF (Fig 7g) could still indicate ligand independent MR1 recognition. Ac-6-FP blocking should be done in this experiment.

In accordance with the reviewer's suggestion, we have added this data in the revised Fig. 6 g (original Fig. 7 g).

The data presented on the other clonotypes, which are also reduced on an MR1^{-/-} background would be important to include in the manuscript.

We thank the reviewer for this constructive comment. Accordingly, we added the description of the other clonotypes that are reduced on an MR1^{-/-} background to the Discussion section of the revised manuscript (page 20, line 12).

The conclusions around RF being the antigen should be modified given the lack of concrete evidence for this.

E.g. Abstract, line 11-13 "...identified the antigen of the TCR expressed by pathogenic T cell..." should be modified. RF is a candidate antigen that may be recognized by one clone (albeit the most frequent). It is also not clear that this clone is responsible for the pathology as several other clonotypes were also observed.

We apologize for our overstated description regarding RF in the original manuscript. We agree with the reviewer's suggestion and have revised our sentences accordingly (page 3, line 11).

Discussion (page 20, line 5): "Given that RF activated T cells in Bcl11bΔiThy mice, this metabolite must regulate MR1-reactive T cells as both an agonist and antagonist depending on the TCR repertoire". There is no evidence for RF activating the cells in the mice (only the presence of the microbiome). There also appears to be no evidence for the latter part of the sentence.

We apologize for our misleading description in the original Discussion. We have

carefully corrected this part of the discussion as per the referee's constructive comment (page 18, line 5).

Minor comments (previous)

4) Sup fig 1B. There is now a major issue with this figure.

To clarify, by "scale" I meant that the same magnification should be used for WT and *Bcl11b* samples. The WT sample in the new Supp figure appears to be the same image from the previous version of the figure, which is now stretched in the horizontal dimension. It is thus unclear if the two samples are of comparable magnification. Please correct.

We deeply apologize for this mistake. During the figure preparation process, the vertical and horizontal ratios were altered inadvertently. This was unintentional as this alteration does not strengthen our conclusion in the presence of clear typical colitis symptoms, such as colon thickness. We carefully confirmed the magnification of original data and have placed the correct data (shown below) in the new supplementary Fig. 1 **b** of the revised manuscript.

Further comments on changes made to the revised manuscript.

(major)

1) According to data shown Fig 1a, the mice did not lose weight as described in the results (Page 6, Line 4). From this data, it would be more correct to say that they did not gain weight like their WT counterparts. However, this data is inconsistent with that shown in Fig 6a where the SPF housed mice gain weight normally, but then lose weight sharply from around 10 weeks of age. Can the authors explain this inconsistency?

We agree with the reviewer's comment and have modified our description in the Results section of our revised manuscript (page 6, line 3). As for the apparent inconsistency between original Fig. 1 **a** and Fig. 6 **a**, we apologize for our misleading presentation. The y-axis in the original Fig. 6 **a** starts from 15, whereas y-axis in Fig. 1 **a** starts from 0. To

avoid the misleading impression that *Bcl11b*^{ΔiThy} mice lost weight sharply around 10 weeks of age, we have set the y-axis in both figures (revised Fig. 1 **a** and Fig. 5 **a**) to start from 0 in the revised manuscript.

2) supp Fig 1d. It is impossible to interpret this figure. What do the symbols represent?

We apologize for the unclear description. Each circle represents an individual mouse, with the corresponding inflammation in the different tissues indicated above the chart in the manner shown previously (Anderson et al., Science, 2002). We have also added a clearer description in the revised supplementary figure legend.

3) Supp Fig 2e. Why are the cells from *Bcl11b*-thy mice only shown for CD4⁺ cells? There is no mention of CD4 in the manuscript prior to this point. Also the Figure legend states that WT cells are shown, but they are not in the figure.

We apologize for not showing this issue in detail. As accumulated T cells in the periphery of *Bcl11b*^{ΔiThy} mice are mostly CD4⁺ T cells (shown below), we used CD4⁺ T cells for adoptive transfer experiments (Supplementary Fig. 2 **b**). We have added this explanation in the Methods section of the revised manuscript (page 29, line 13). In addition, we have clearly labeled the genotypes of mice in Supp Fig 2 **e** of the revised manuscript.

4) Figure 2h. Page 9, line 15: text describes expansion of *Bcl11b*^{ΔiThy} cells, figure shows CD4 T/B cells (%), but the use of CD4 here is not explained. Also CD4 T/B cells (%) does not make sense.

We apologize for the lack of description and unclear presentation in Fig. 2 **h**. As we mentioned above, we analyzed CD4⁺ T cells because the CD4⁺ population in *Bcl11b*^{ΔiThy} mice had a potential to cause autoimmune disease (Supplementary Fig. 2 **b**). We now explain this in the Methods section (page 29, line 12). We have also changed the

presentation of the data to show CD4⁺ αβ T cell frequencies among whole PBMCs, but not the ratio to B cells, in the revised Fig. 2 **h** as shown below.

Page 9, line 15 “...partly due to the survival disadvantages.” This is not shown. If it is speculation that this may be the cause this should be stated as a possible cause not definitively as written here.

In our previous response letter to the reviewer #2, we showed a data suggesting that *Bcl11b*^{ΔiThy} T cells were prone to apoptosis (shown below). However, in accordance with the reviewer’s suggestion, we have carefully corrected this description in the revised manuscript (page 9, line 16).

5) Page 9, Line 16: chronic inflammation induced by *Bcl11b*^{ΔiThy} BM cells was eliminated by co-transfer with WT BM cells (Fig 2k, Supp Fig 2f). The supp figure shows only one representative image for each. Fig 2k appears to show 2 mice, but there is no statistical test shown. Was this significant?.

The images we show in Fig 2 **k** and Supp Fig 2 **f** are representative of at least three independent mice with similar inflammatory symptoms. We apologize for not performing a statistical test in Fig. 2 **k**. Differences are significant if we combine experimental data

from two independent experiments as shown below. Therefore, we have added this data in the revised manuscript as revised Fig. 2 k.

6) Apologies for not picking this up earlier, but there is an extra residue in the CDRa (Fig 3f) compared to typical MAIT TCRs. Critically, this would shift the Tyr95 residue (thus no longer at position 95), impacting interpretation of the crystal structure virtual superposition. Indeed, Supp Fig 3e clearly shows a different position for the Tyr, while the authors' description that it points towards the cleft remains true. Thus, the text should be modified to accommodate this finding, which is actually very interesting.

We thank the reviewer for their careful analysis in raising this point. In accordance with the reviewer's suggestion, we have mentioned this point in the revised manuscript (page 11, line 7).

(minor)

Abstract. Line 8-9 "a skewed receptor" or "skewed receptors"

We thank the reviewer for this comment. We have corrected the mistake in the revised manuscript (page 3, line 8).

Line 10. "...type 2 MAIT cells have not been reported". There have been reports of type 2 cytokine production by MAIT cells, so this statement should be adjusted.

We apologize for the incorrect description. We have changed the sentences and added the appropriate reference in the revised manuscript (page 4, lines 10 / pages 20, line 1).

Abstract. Final sentence is not needed as the result was stated above regarding germ-free

conditions.

Page 6, line 8. "As" is not needed in this sentence.

In accordance with the reviewer's suggestion, we have corrected the redundant sentences in the Abstract (page 3, line 13) and the Results section (page 6, line 8) of the revised manuscript.

Page 10, line 2: Fig 2l and m. Only 3-4 mice per group is insufficient. There is no mention of repeat experiments for l. In m, a paired test could be used for the donor/recipient cells within the same mouse.

We apologize for the lack of description. We have now added the description of Fig. 2 **l** experimental repeats to the Figure legends section of the revised manuscript (page 56, line 6). In the revised Fig. 2 **l** and **m**, we have shown combined experimental data and accordingly have corrected the description (page 55, line 15). Following the reviewer's constructive suggestion, we have determined the statistical significance by a paired-t test within the same mouse in Fig. 2 **m**.

Reviewer #2 (Remarks to the Author):

All of my comments were sufficiently addressed.

Reviewer #3 (Remarks to the Author):

This paper by Shibata et al. provided interesting study on non-MAIT MR1-restricted T cells, but there are issues that must be addressed to draw conclusions regarding the function and contribution of these cells in autoinflammatory disorders.

1. The authors claim that MR1-restricted T cells including clonotype #1 T cells are responsible for pancreatitis because they are accumulated in the pancreas. However, they only showed that “CD4+ cells” observed in the pancreas of *Bcl11b* Δ iThy mice were disappeared by MR1 deficiency in these mice (Figure 4d). The pathology in *Bcl11b* Δ iThy mice seems dependent on MR1 and T cells, but data presented here do not demonstrate that MR1-restricted clonotype #1 T cells are actually responsible for the disease in *Bcl11b* Δ iThy mice unless the dependency is confirmed by experiments with as the adaptive transfer of these cells or agonistic agents for these cells, or by using mice deficient with this specific T cell population. The current data in this paper suggest that any T cells restricted by MR1 may be responsible for the disease in *Bcl11b* Δ iThy mice. It is well recognized that MR1 deficiency influences microbial composition (Varelias A JCI 2019; Smith AD PLOS ONE 2019), and disease phenotype in MR1^{+/+} mice co-housed with MR1^{-/-} mice is often different from that in wild-type mice obtained from a vendor. Thus, to confirm the contribution of MR1-restricted T cells, *Bcl11b* Δ iThy x *Mr1*^{-/-} mice should be compared with co-housed littermate *Bcl11b* Δ iThy mice (Fig 2d, 2e, 2f, 3a-j, 4d-f). To demonstrate the contribution of MR1-restricted T cells in the pathogenesis, *Bcl11b* Δ iThy x *Mr1*^{-/-} mice (as well as co-housed littermate *Bcl11b* Δ iThy mice) should be also analyzed for experiments in Fig. 4c, 4g-j, 4k, 4l, 6 a-k).

We agree to the reviewer's comment that we should continuously work to show that MR1-restricted clonotype #1 T cells are actually responsible for the disease in *Bcl11b* Δ iThy mice. Currently, adaptive transfer of these clonotypes can not be addressed as clonotype-specific antibodies/tetramers are not available. The administration of RF did not induce inflammation in WT mice under T-balanced conditions (data not shown). Instead, in addition to MR1-deficient mice, we attempted to address this question from a different perspective by generating *Tra33*-deficient mice and crossing them with *Bcl11b* Δ iThy mice. As shown in revised Fig. 3 h, *Bcl11b* Δ iThy mice experienced prolonged survival when *Tra33*⁺ cells were absent (shown below), providing an additional implication that these cells contribute to the pathogenesis.

In accordance with the reviewer’s suggestion, we performed metagenomic analyses of WT, *Mr1*^{-/-} and *Traj33*^{-/-} mice in our colonies. As shown below (left), a slight difference in the composition of microbiota was detected in *Mr1*^{-/-} mice as reported (Smith et al 2019); however, no significant change was observed as the microbiota composition of both WT and *Traj33*^{-/-} mice displayed comparable levels of Chao1 richness estimation index values of alpha diversity (shown below, right) (Smith et al., PLoS One, 2019). Thus, it is unlikely that the improvement of inflammatory symptoms observed in *Traj33*-deficient *Bcl11b*^{ΔiThy} mice is caused by changes of microbiota.

2. The authors demonstrated inflammation in the pancreas and colitis and hypergammaglobulinemia (including IgM, IgG, IgA, and IgE) in *Bcl11b*^{ΔiThy} mice. However, in human IgG4-related disease, the colitis is very rare (0.8%) and the exclusion criteria includes comorbidities including inflammatory bowel diseases (2019 ACR/EULAR Classification criteria for IgG4-related disease). In addition, there is no elevation of IgM and

IgA in IgG4-related disease (Rheumatology 2015. 54: 1982–1990). Thus, the disease in observed in Bcl11b Δ iThy mice should be considered different from IgG4-related disease and Figure 5 should be excluded from the paper. I think Figure 5 is not informative because age and sex of healthy and AIP specimens appear to be not matched. In addition, non-MAIT MR1-restricted T cells were identified based on the recognition of Ac-6-Fc which is not recognized by clonotype #1 T cells.

In accordance with the reviewer's recommendation, we have deleted human data in the original Fig. 5.

REVIEWER COMMENTS

Reviewer #1 (Remarks to the Author):

All comments have been addressed.

Reviewer #3 (Remarks to the Author):

The authors have addressed one of the concerns by excluding the figure 5, but other major concerns remain unaddressed. Thus, the presented data are insufficient to draw their conclusions.

One of the remaining concerns was that the proper control co-housed littermate Bcl11b Δ iThy mice are missing in animal experiments using Bcl11b Δ iThy x Mr1 $^{-/-}$ mice in Figure 2, 3, and 4. Sorry for not being able to point out this critical issue earlier because the methods and figure legends were not well described in the original manuscript.

In general, littermate controls are needed for animal experiments of disease models. The hybrid strains between 129 and CD57L/6 mice are widely used in knockout mice including MR1-deficient mice and other mouse strains used in this study. It has been shown that the crossing 129 and CD57L/6 mouse strains resulted spontaneous lupus-like disease by creating multiple genetic loci contributing to autoimmunity. Thus, littermate controls are indispensable especially when studying autoimmunity using such genetically engineered mice.

Another reason that co-housed littermate control mice are required in experiments using Mr1 $^{-/-}$ mice that MR1-restricted T cells recognize antigens derived from commensal microbiota, and this might have effects on the pathogenesis of disease models. The authors claim that they analyzed the microbiota in WT, Mr1 $^{-/-}$ as well as Traj3 $^{-/-}$ mice "as suggested by the reviewer" (although I did not), and concluded that WT mice instead of littermate controls can be used as controls because the Chao1 index values of microbiota in these mice strains are comparable. I disagree with their point because Chao 1 index only estimates the diversity of microbiota composition, and this result does not indicate that these strains of mice have similar microbiota. Given the strong interindividual differences seen in Figure 2e, 4d, and 4f, analysis using littermate controls are required in Fig 2d, 2e, 2f, 3a-j, 4d-f.

Another questions related figure 2e are why the number of Bcl11b Δ iThy mice is greater than that of Bcl11b Δ iThy Mr1 $^{-/-}$ mice even though the survival rate is much lower in Bcl11b Δ iThy mice, and how possible to obtain 6 Bcl11b Δ iThy mice at 15 weeks of age even though the majority of 12 mice die by that time. All mice survived in both groups should be analyzed, and this is also applied to Figure 4d, 4f, 4g, 5e, and 5g.

The authors included additional data of Traj33 $^{-/-}$ Bcl11b Δ iThy mice to support their conclusion that Trv7-6-Traj33/Trbv13-2-Trbj1-1 T cells (clonotype #1) play a major pathogenic role in Bcl11b Δ iThy mice. However, it is not shown how many of mice were analyzed and that these Traj33 $^{-/-}$ cells are restricted MR1 and actually represent clonotype #1, and again, co-housed littermate controls for Traj3 $^{-/-}$ Bcl11b Δ iThy mice are not included in the experiments.

Reviewer #3 (Remarks to the Author):

The authors have addressed one of the concerns by excluding the figure 5, but other major concerns remain unaddressed. Thus, the presented data are insufficient to draw their conclusions.

One of the remaining concerns was that the proper control co-housed littermate Bcl11b Δ iThy mice are missing in animal experiments using Bcl11b Δ iThy x Mr1 $^{-/-}$ mice in Figure 2, 3, and 4. Sorry for not being able to point out this critical issue earlier because the methods and figure legends were not well described in the original manuscript.

In general, littermate controls are needed for animal experiments of disease models. The hybrid strains between 129 and CD57L/6 mice are widely used in knockout mice including MR1-deficient mice and other mouse strains used in this study. It has been shown that the crossing 129 and CD57L/6 mouse strains resulted spontaneous lupus-like disease by creating multiple genetic loci contributing to autoimmunity. Thus, littermate controls are indispensable especially when studying autoimmunity using such genetically engineered mice.

Another reason that co-housed littermate control mice are required in experiments using Mr1 $^{-/-}$ mice that MR1-restricted T cells recognize antigens derived from commensal microbiota, and this might have effects on the pathogenesis of disease models. The authors claim that they analyzed the microbiota in in WT, Mr1 $^{-/-}$ as well as Traj3 $^{-/-}$ mice “as suggested by the reviewer” (although I did not), and concluded that WT mice instead of littermate controls can be used as controls because the Chao1 index values of microbiota in these mice strains are comparable. I disagree with their point because Chao 1 index only estimates the diversity of microbiota composition, and this result does not indicate that these strains of mice have similar microbiota. Given the strong interindividual differences seen in Figure 2e, 4d, and 4f, analysis using littermate controls are required in Fig 2d, 2e, 2f, 3a-j, 4d-f.

Another questions related figure 2e are why the number of Bcl11b Δ iThy mice is greater than that of Bcl11b Δ iThy Mr1 $^{-/-}$ mice even though the survival rate is much lower in Bcl11b Δ iThy mice, and how possible to obtain 6 Bcl11b Δ iThy mice at 15 weeks of age even though the majority of 12 mice die by that time. All mice survived in both groups should be analyzed, and this is also applied to Figure 4d, 4f, 4g, 5e, and 5g.

The authors included additional data of Traj33^{-/-} Bcl11b^{ΔiThy} mice to support their conclusion that Trav7-6-Traj33/Trbv13-2-Trbj1-1 T cells (clonotype #1) play a major pathogenic role in Bcl11b^{ΔiThy} mice. However, it is not shown how many of mice were analyzed and that these Traj33^{-/-} cells are restricted MR1 and actually represent clonotype #1, and again, co-housed littermate controls for Traj3^{-/-} Bcl11b^{ΔiThy} mice are not included in the experiments.

We apologize that we did not clearly describe the control mice used in this study. We mostly used littermate mice for control groups and designated Rag1⁻, Bcl11b⁻ and MR1-sufficient mice (*Rag1*^{Cre/+ or +/+} × *Bcl11b*^{+/+} × *Mr1*^{+/-}, *Rag1*^{Cre/+ or +/+} × *Bcl11b*^{flox/+} × *Mr1*^{+/-}, *Rag1*^{+/+} × *Bcl11b*^{flox/flox or flox/+} × *Mr1*^{+/+}, *Rag1*^{+/+} × *Bcl11b*^{flox/flox} × *Mr1*^{+/+ or +/-}, *Rag1*^{+/+} × *Bcl11b*^{flox/+} × *Mr1*^{+/+} mice) as WT mice in the Figures pointed out by reviewer #3: Fig. 2 d-f, Fig. 3 a-d and Fig. 4 d-f (shown below). Some other control mice were not co-housed littermates, but all control mice used in these figures were born and bred in the same room with the MR1-deficient mice (shown below). We have added this information to the Methods section of the revised manuscript (page 24, line 10).

Fig. 2 d

Cage #	Date of birth	Gender	Rag1	Bcl11b	Mr1	Description in the Figure legends	
1	2015.6.12	♂	Cre/+	flox/flox	-/-	Bcl11b ^{ΔiThy} × Mr1 ^{-/-}	Littermate pairs
2	2015.6.18	♂	Cre/+	flox/flox	+/-	Bcl11b ^{ΔiThy}	
2	2015.6.18	♂	Cre/+	flox/flox	+/-	Bcl11b ^{ΔiThy}	Littermate pairs
3	2015.6.20	♂	Cre/+	flox/flox	+/-	Bcl11b ^{ΔiThy}	
3	2015.6.20	♂	Cre/+	flox/flox	+/-	Bcl11b ^{ΔiThy}	Littermate pairs
4	2015.8.29	♂	Cre/+	flox/flox	+/+	Bcl11b ^{ΔiThy}	
5	2015.9.2	♀	Cre/+	flox/flox	-/-	Bcl11b ^{ΔiThy} × Mr1 ^{-/-}	Littermate pairs
6	2015.9.11	♂	Cre/+	flox/flox	+/-	Bcl11b ^{ΔiThy}	
6	2015.9.11	♂	Cre/+	flox/flox	+/-	Bcl11b ^{ΔiThy}	Littermate pairs
6	2015.9.11	♂	Cre/+	flox/flox	+/+	Bcl11b ^{ΔiThy}	
6	2015.9.11	♂	Cre/+	flox/flox	+/+	Bcl11b ^{ΔiThy}	Littermate pairs
7	2015.9.11	♀	Cre/+	flox/flox	-/-	Bcl11b ^{ΔiThy} × Mr1 ^{-/-}	
7	2015.9.11	♀	Cre/+	flox/flox	+/+	Bcl11b ^{ΔiThy}	Littermate pairs
8	2015.9.18	♂	Cre/+	flox/flox	-/-	Bcl11b ^{ΔiThy} × Mr1 ^{-/-}	
8	2015.9.18	♂	Cre/+	flox/flox	+/-	Bcl11b ^{ΔiThy}	Littermate pairs
9	2015.9.23	♀	Cre/+	flox/flox	-/-	Bcl11b ^{ΔiThy} × Mr1 ^{-/-}	
9	2015.9.23	♀	Cre/+	flox/flox	+/+	Bcl11b ^{ΔiThy}	Littermate pairs
10	2015.10.13	♀	Cre/+	flox/flox	-/-	Bcl11b ^{ΔiThy} × Mr1 ^{-/-}	
10	2015.10.13	♀	Cre/+	flox/flox	-/-	Bcl11b ^{ΔiThy} × Mr1 ^{-/-}	Littermate pairs
10	2015.10.14	♀	Cre/+	flox/flox	-/-	Bcl11b ^{ΔiThy} × Mr1 ^{-/-}	
10	2015.10.14	♀	Cre/+	flox/flox	-/-	Bcl11b ^{ΔiThy} × Mr1 ^{-/-}	Littermate pairs

Fig. 2 e

Cage #	Date of birth	Gender	Rag1	Bcl11b	Mr1	Description in the Figure legends
1	2015.9.18	♂	Cre/+	flox/flox	-/-	Bcl11b ^{ΔTby} × Mr1 ^{-/-}
2	2015.9.23	♀	Cre/+	flox/flox	-/-	Bcl11b ^{ΔTby} × Mr1 ^{-/-}
3	2015.10.14	♀	Cre/+	flox/flox	-/-	Bcl11b ^{ΔTby} × Mr1 ^{-/-}
3	2015.10.14	♀	Cre/+	flox/flox	-/-	Bcl11b ^{ΔTby} × Mr1 ^{-/-}
4	2015.11.9	♀	Cre/+	flox/flox	+/-	Bcl11b ^{ΔTby}
5	2015.11.18	♀	Cre/+	flox/flox	+/-	Bcl11b ^{ΔTby}
6	2016.5.1	♀	Cre/+	flox/flox	+/+	Bcl11b ^{ΔTby}
7	2018.1.20	♀	Cre/+	flox/flox	+/-	Bcl11b ^{ΔTby}
7	2018.1.20	♀	Cre/+	flox/flox	+/-	Bcl11b ^{ΔTby}
7	2018.1.20	♀	Cre/+	flox/flox	+/-	Bcl11b ^{ΔTby}

Littermate pairs

Littermate pairs

Fig. 2 f

Cage #	Date of birth	Gender	Rag1	Bcl11b	Mr1	Description in the Figure legends
1	2019.11.9	♂	Cre/+	+/+	+/-	WT
2	2019.11.9	♀	Cre/+	flox/flox	+/-	Bcl11b ^{ΔTby}
2	2019.11.9	♀	Cre/+	flox/flox	-/-	Bcl11b ^{ΔTby} × Mr1 ^{-/-}
3	2019.12.3	♂	Cre/+	flox/flox	-/-	Bcl11b ^{ΔTby} × Mr1 ^{-/-}
4	2019.12.26	♂	Cre/+	flox/flox	+/-	Bcl11b ^{ΔTby}
5	2020.1.9	♀	+/+	+/+	+/+	WT
6	2020.7.17	♀	+/+	flox/flox	+/+	WT
7	2020.7.17	♂	Cre/+	flox/flox	+/+	Bcl11b ^{ΔTby}
7	2020.7.17	♂	Cre/+	flox/flox	-/-	Bcl11b ^{ΔTby} × Mr1 ^{-/-}

Littermate pairs

Littermate pairs

Fig. 3 a-d

Cage #	Date of birth	Gender	Rag1	Bcl11b	Mr1	Description in the Figure legends
1	2019.11.9	♀	Cre/+	flox/flox	+/-	Bcl11b ^{ΔTby}
1	2019.11.9	♀	Cre/+	flox/flox	-/-	Bcl11b ^{ΔTby} × Mr1 ^{-/-}

Littermate pairs

Fig. 4 d

Cage #	Date of birth	Gender	Rag1	Bcl11b	Mr1	Description in the Figure legends
1	2015.9.11	♂	+/+	flox/+	+/-	WT
1	2015.9.11	♂	+/+	flox/+	+/-	WT
1	2015.9.11	♂	+/+	flox/flox	+/+	WT
1	2015.9.11	♂	+/+	flox/+	+/+	WT
2	2015.9.18	♂	Cre/+	flox/flox	-/-	Bcl11b ^{ΔTby} × Mr1 ^{-/-}
3	2015.9.23	♀	Cre/+	flox/flox	-/-	Bcl11b ^{ΔTby} × Mr1 ^{-/-}
4	2015.10.14	♀	Cre/+	flox/flox	-/-	Bcl11b ^{ΔTby} × Mr1 ^{-/-}
4	2015.10.14	♀	Cre/+	flox/flox	-/-	Bcl11b ^{ΔTby} × Mr1 ^{-/-}
5	2015.11.9	♀	Cre/+	flox/flox	+/-	Bcl11b ^{ΔTby}
6	2015.11.18	♀	Cre/+	flox/flox	+/-	Bcl11b ^{ΔTby}
7	2016.5.1	♀	Cre/+	flox/flox	+/+	Bcl11b ^{ΔTby}
8	2018.1.20	♀	Cre/+	flox/flox	+/-	Bcl11b ^{ΔTby}

Littermate pairs

Littermate pairs

Fig. 4 e

Cage #	Date of birth	Gender	Rag1	Bcl11b	Mr1	Description in the Figure legends
1	2015.9.11	♂	+/+	flox/+	+/-	WT
1	2015.9.11	♂	+/+	flox/+	+/-	WT
1	2015.9.11	♂	+/+	flox/flox	+/+	WT
1	2015.9.11	♂	+/+	flox/+	+/+	WT
2	2015.9.11	♀	+/+	flox/+	+/-	WT
1	2015.9.11	♂	Cre/+	flox/flox	+/-	Bcl11b ^{ΔTby}
1	2015.9.11	♂	Cre/+	flox/flox	+/-	Bcl11b ^{ΔTby}
1	2015.9.11	♂	Cre/+	flox/flox	+/+	Bcl11b ^{ΔTby}
1	2015.9.11	♂	Cre/+	flox/flox	+/+	Bcl11b ^{ΔTby}
2	2015.9.11	♀	Cre/+	flox/flox	+/+	Bcl11b ^{ΔTby}
2	2015.9.11	♀	Cre/+	flox/flox	-/-	Bcl11b ^{ΔTby} × Mr1 ^{-/-}
3	2015.9.18	♂	Cre/+	flox/flox	+/-	Bcl11b ^{ΔTby}
3	2015.9.18	♂	Cre/+	flox/flox	-/-	Bcl11b ^{ΔTby} × Mr1 ^{-/-}
4	2015.9.23	♀	+/+	flox/flox	+/-	WT
4	2015.9.23	♀	Cre/+	flox/flox	+/+	Bcl11b ^{ΔTby}
4	2015.9.23	♀	Cre/+	flox/flox	-/-	Bcl11b ^{ΔTby} × Mr1 ^{-/-}

Fig. 4 f

Cage #	Date of birth	Gender	Rag1	Bcl11b	Mr1	Description in the Figure legends
1	2015.6.18	♂	Cre/+	flox/+	+/-	WT
1	2015.6.18	♂	+/+	+/+	+/-	WT
1	2015.6.18	♂	+/+	flox/+	+/-	WT
2	2015.8.29	♀	+/+	flox/+	+/+	WT
3	2015.9.11	♂	+/+	flox/+	+/-	WT
3	2015.9.11	♂	+/+	flox/+	+/-	WT
3	2015.9.11	♂	+/+	flox/flox	+/+	WT
3	2015.9.11	♂	+/+	flox/+	+/+	WT
4	2015.9.11	♀	+/+	flox/+	+/-	WT
4	2015.9.11	♀	Cre/+	flox/flox	-/-	Bcl11b ^{ΔTby} × Mr1 ^{-/-}
5	2015.9.18	♂	Cre/+	flox/flox	+/-	Bcl11b ^{ΔTby}
5	2015.9.18	♂	Cre/+	flox/flox	-/-	Bcl11b ^{ΔTby} × Mr1 ^{-/-}
6	2015.9.23	♀	Cre/+	flox/flox	-/-	Bcl11b ^{ΔTby} × Mr1 ^{-/-}
7	2015.10.14	♀	Cre/+	flox/flox	-/-	Bcl11b ^{ΔTby} × Mr1 ^{-/-}
7	2015.10.14	♀	Cre/+	flox/flox	-/-	Bcl11b ^{ΔTby} × Mr1 ^{-/-}
8	2015.11.9	♀	Cre/+	flox/flox	+/-	Bcl11b ^{ΔTby}
9	2015.11.18	♀	Cre/+	flox/flox	+/-	Bcl11b ^{ΔTby}
10	2016.5.1	♀	Cre/+	flox/flox	+/+	Bcl11b ^{ΔTby}
11	2018.1.20	♀	Cre/+	flox/flox	+/-	Bcl11b ^{ΔTby}
11	2018.1.20	♀	Cre/+	flox/flox	+/-	Bcl11b ^{ΔTby}
11	2018.1.20	♀	Cre/+	flox/flox	+/-	Bcl11b ^{ΔTby}

MR1-deficient mice used in this study were originally developed by Dr. Susan Gilfillan on an F1 hybrid background of 129 and C57BL/6 strains (Treiner, et al, *Nature*, 2003), but they have been backcrossed to C57BL/6 mice at least 20 generations (10 generations as described previously (Croxford, et al., *Nat. Immunol.*, 2006) followed by further backcrossing of more than 10 generations to C57BL/6 mice by our study). We have added this information to the Methods section of the revised manuscript (page 24, line 8). As significant differences were detected in the experiments shown in Fig. 2 d and e, Fig. 4 d-f, we could not conclude that these differences are due to the background of mouse strains or different microbiota compositions.

In accordance with the reviewer's request, in addition to Chao1 values used in the

indicated paper (Smith, et al., *PLoS One*, 2019), we have measured beta diversity of fecal bacteria (weighted and unweighted UniFrac distances) between WT and *Mr1*^{-/-} mice using the analysis of similarity (ANOSIM) test (shown below). There was no significant difference (*p*-value > 0.05) between them, suggesting that deletion of the *Mr1* gene did not cause drastic changes in the composition of microbiota, at least in our colony. From these additional experiments, we could not prove that a possible difference of microbiota was the cause of the different clinical symptoms between *Bcl11b*^{ΔiThy} mice and *Bcl11b*^{ΔiThy} × *Mr1*^{-/-} mice.

Although the majority of *Bcl11b*^{ΔiThy} mice died by 12 weeks of age as shown in Fig. 2 d, as the reviewer pointed out, some survived longer. As we designed the experiment in Fig. 2 d by mainly focusing on mortality, it was not possible to use all surviving mice in Fig. 2 d for the analysis in Fig. 4 d, f and g, Fig. 5 e and g. As *Bcl11b*^{ΔiThy} × *Mr1*^{-/-} mice that survived beyond 15 weeks of age were also used for other experiments, we analyzed four mice for the experiment in Fig. 2 e.

We apologize that the number of *Bcl11b*^{ΔiThy} × *Traj33*^{-/-} mice (n = 5) was described only in the original Fig. 3 h legend. Most of MR1-restricted T cells are Jα33⁺ T cells, as MR1-restricted T cells were virtually absent in *Traj33*^{-/-} mice (Koay, et al., *Nat. Commun.* 2019). Thus, we newly generated *Traj33*^{-/-} mice and crossed them with *Bcl11b*^{ΔiThy} mice to delete clonotype #1 (Jα33⁺), since no probes specifically detecting/deleting only clonotype #1 are available to date. Although this model is currently the only available approach to eliminate clonotype #1 *in vivo* (except for an MR1-deficient background),

$\alpha 33^+$ T cells other than clonotype #1 are also missing in *Tra33*^{-/-} mice and we cannot fully exclude the contribution of these cells to the pathogenesis of *Bcl11b* ^{Δ Thy} mice. Furthermore, as the reviewer pointed out, we did not use littermate mice as control groups for *Bcl11b* ^{Δ Thy} \times *Tra33*^{-/-} mice as shown below. Considering these limitations, we have decided to delete this data from the revised manuscript.

Original Fig. 3 h

Cage #	Date of birth	Gender	Rag1	Bcl11b	Tra33	Description in the Figure legends
1	2020.3.7	♀	Cre/+	flox/flox	+/+	Bcl11b ^{ΔThy}
1	2020.3.7	♀	Cre/+	flox/flox	+/+	Bcl11b ^{ΔThy}
1	2020.3.7	♀	Cre/+	flox/flox	+/+	Bcl11b ^{ΔThy}
1	2020.3.7	♀	Cre/+	flox/flox	+/+	Bcl11b ^{ΔThy}
2	2020.3.9	♀	Cre/+	flox/flox	+/+	Bcl11b ^{ΔThy}
2	2020.3.9	♀	Cre/+	flox/flox	+/+	Bcl11b ^{ΔThy}
2	2020.3.9	♀	Cre/+	flox/flox	+/+	Bcl11b ^{ΔThy}
3	2020.4.22	♂	Cre/+	flox/flox	+/+	Bcl11b ^{ΔThy}
3	2020.4.22	♂	Cre/+	flox/flox	+/+	Bcl11b ^{ΔThy}
4	2020.5.11	♂	Cre/+	flox/flox	+/+	Bcl11b ^{ΔThy}
4	2020.5.11	♂	Cre/+	flox/flox	+/+	Bcl11b ^{ΔThy}
4	2020.5.11	♂	Cre/+	flox/flox	+/+	Bcl11b ^{ΔThy}
4	2020.5.11	♂	Cre/+	flox/flox	+/+	Bcl11b ^{ΔThy}
5	2020.5.11	♀	Cre/+	flox/flox	+/+	Bcl11b ^{ΔThy}
5	2020.5.11	♀	Cre/+	flox/flox	+/+	Bcl11b ^{ΔThy}
5	2020.5.11	♀	Cre/+	flox/flox	+/+	Bcl11b ^{ΔThy}
6	2020.6.30	♀	Cre/+	flox/flox	+/+	Bcl11b ^{ΔThy}
6	2020.6.30	♀	Cre/+	flox/flox	+/+	Bcl11b ^{ΔThy}
7	2020.9.26	♀	Cre/+	flox/flox	+/+	Bcl11b ^{ΔThy}
7	2020.9.26	♀	Cre/+	flox/flox	+/+	Bcl11b ^{ΔThy}
7	2020.9.26	♀	Cre/+	flox/flox	+/+	Bcl11b ^{ΔThy}
8	2020.10.5	♀	Cre/+	flox/flox	+/+	Bcl11b ^{ΔThy}
8	2020.10.5	♀	Cre/+	flox/flox	+/+	Bcl11b ^{ΔThy}
9	2020.12.7	♂	Cre/+	flox/flox	+/+	Bcl11b ^{ΔThy}
9	2020.12.7	♂	Cre/+	flox/flox	+/+	Bcl11b ^{ΔThy}
10	2021.1.4	♀	Cre/+	flox/flox	+/+	Bcl11b ^{ΔThy}
11	2021.1.10	♂	Cre/+	flox/flox	+/+	Bcl11b ^{ΔThy}
12	2021.6.3	♂	Cre/+	flox/flox	-/-	Bcl11b ^{ΔThy} \times Tra33 ^{+/-}
12	2021.6.3	♂	Cre/+	flox/flox	-/-	Bcl11b ^{ΔThy} \times Tra33 ^{+/-}
13	2021.6.3	♀	Cre/+	flox/flox	-/-	Bcl11b ^{ΔThy} \times Tra33 ^{+/-}
14	2021.9.6	♀	Cre/+	flox/flox	-/-	Bcl11b ^{ΔThy} \times Tra33 ^{+/-}
14	2021.9.6	♀	Cre/+	flox/flox	-/-	Bcl11b ^{ΔThy} \times Tra33 ^{+/-}

REVIEWERS' COMMENTS

Reviewer #3 (Remarks to the Author):

The standard method for experiments with genetically modified strains of mice is to use F2-generation littermates born on the same date from the same breeding pair of heterozygous parents. Assuming the mice born on the same birthdate are co-housed littermates, the numbers of *Bcl11b* Δ *Thy* x *Mr1*^{-/-} mice vs their littermate controls are 3 vs 7 (Figure 2d); none (Figure 2e); 2 vs 4 (Figure 2f); 1 vs 1 (Figure 3); none (Figure 4d); 3 vs 7 (Figure 4e); and 1 vs 1 (Figure 4f). Apparently, any conclusions cannot be drawn based on these data with so little numbers (or none) of proper control mice. As the reviewer #1 had also pointed out, each group should have at least 6-8 mice, and even more mice where there are interindividual differences as seen in Figure 2e, 4d, and 4f. The numbers of male and female mice should be also matched among groups, and each animal experiment should be repeated 2-3 times.

Regarding the microbiota analysis, the authors added these data which they claim they added upon reviewer's request, but they were not suggested by reviewers. Again, they claim that they performed additional analysis on microbiota responding to the reviewer's request, but these data are not required.

Reviewer #3 (Remarks to the Author):

The standard method for experiments with genetically modified strains of mice is to use F2-generation littermates born on the same date from the same breeding pair of heterozygous parents. Assuming the mice born on the same birthdate are co-housed littermates, the numbers of *Bcl11b* Δ iThy x *Mr1*^{-/-} mice vs their littermate controls are 3 vs 7 (Figure 2d); none (Figure 2e); 2 vs 4 (Figure 2f); 1 vs 1 (Figure 3); none (Figure 4d); 3 vs 7 (Figure 4e); and 1 vs 1 (Figure 4f). Apparently, any conclusions cannot be drawn based on these data with so little numbers (or none) of proper control mice. As the reviewer #1 had also pointed out, each group should have at least 6-8 mice, and even more mice where there are interindividual differences as seen in Figure 2e, 4d, and 4f. The numbers of male and female mice should be also matched among groups, and each animal experiment should be repeated 2-3 times.

We thank the reviewer for raising important criticisms. To minimize interindividual genetic differences between *Bcl11b*^{ΔiThy} mice and *Bcl11b*^{ΔiThy} × *Mr1*^{-/-} mice, we used *Mr1*^{-/-} mice backcrossed to C57BL/6 mice more than 20 generations. We have also shown in the response letter after 2nd revision that deletion of the *Mr1* gene did not cause drastic changes in the composition of microbiota in our colony. These results suggest the unlikelihood of our *in vivo* experimental design, rather than genotypes, as a principal cause of the phenotypes in the current study. We were interested in the sex difference of the disease symptoms, but we did not find substantial difference in survival and pancreatitis/colitis.

Regarding the microbiota analysis, the authors added these data which they claim they added upon reviewer's request, but they were not suggested by reviewers. Again, they claim that they performed additional analysis on microbiota responding to the reviewer's request, but these data are not required.

We followed the reviewer's suggestion.